# TANGENTIAL WASSERSTEIN PROJECTIONS

## ABSTRACT

We develop a notion of projections between sets of probability measures using the geometric properties of the 2-Wasserstein space. In contrast to existing methods, it is designed for multivariate probability measures that need not be regular, is computationally efficient to implement via a linear regression, and provides a unique solution in general. The idea is to work on tangent cones of the Wasserstein space using generalized geodesics. Its structure and computational properties make the method applicable in a variety of settings where probability measures need not be regular, from causal inference to the analysis of object data. An application to estimating causal effects yields a generalization of the synthetic controls method for systems with general heterogeneity described via multivariate probability measures, something that has been out of reach of existing approaches.

## 1 INTRODUCTION

The concept of projections, that is, approximating a target quantity of interest by an optimally weighted combination of other quantities, is of fundamental relevance in learning theory and statistics. Projections are generally defined between random variables in appropriately defined linear spaces (e.g. van der Vaart, 2000, chapter 11). In modern statistics and machine learning applications, the objects of interest are often probability measures themselves. Examples range from object- and functional data (e.g. Marron & Alonso, 2014) to causal inference with individual heterogeneity (e.g. Athey & Imbens, 2015).

A notion of projection between sets of probability measures should be applicable between any set of general probability measures, replicate geometric properties of the target measure, and possess good computational and statistical properties. We introduce such a notion of projection between sets of general probability measures supported on Euclidean spaces. It provides a unique solution to the projection problem under mild conditions. To achieve this, we work in the 2-Wasserstein space, that is, the set of all probability measures with finite second moments equipped with the 2-Wasserstein distance.

Importantly, we focus on the multivariate setting, i.e. we consider the Wasserstein space over some Euclidean space $\mathbb{R}^d$, denoted by $\mathcal{W}_2$, where the dimension $d$ can be high. The multivariate setting poses challenges from a mathematical, computational, and statistical perspective. In particular, $\mathcal{W}_2$ is a positively curved metric space for $d > 1$ (e.g. Ambrosio et al., 2008, Kloeckner, 2010). Moreover, the 2-Wasserstein distance between two probability measures is defined as the value function of the Monge-Kantorovich optimal transportation problem (Villani, 2003, chapter 2), which does not have a closed-form solution in multivariate settings. This is coupled with a well-known statistical curse of dimensionality for general measures (Ajtai et al., 1984, Dudley, 1969, Fournier & Guillin, 2015, Talagrand, 1992; 1994, Weed & Bach, 2019).

### 1.1 EXISTING APPROACHES

These challenges have impeded the development of a method of projections between general and potentially high-dimensional probability measures. A focus so far has been on the univariate and low-dimensional setting. In particular, Chen et al. (2021), Ghodrati & Panaretos (2022), and Pegoraro & Beraha (2021) introduced frameworks for distribution-on-distribution regressions in the univariate setting for object data. Bigot et al. (2014), Cazelles et al. (2017) developed principal component analyses on the space of univariate probability measures using geodesics on the Wasserstein space.

The most closely related works to ours are Bonneel et al. (2016), Mérigot et al. (2020), and Werenski et al. (2022). The first develops a regression approach in barycentric coordinates with applications in computer graphics as well as color and shape transport problems. Their method is defined directly on $\mathcal{W}_2$ and requires solving a computationally costly bilevel optimization problem, which does not necessarily yield global solutions. The second introduces a linearization of the 2-Wasserstein space by lifting it to a $L^2$-space anchored at measure that is absolutely continuous with respect to Lebesgue measure. This approach relies on the existence of optimal transport maps between this absolutely continuous "anchor" distribution and other distributions and hence only defines tangent spaces at absolutely continuous measures. The third works on a tangential structure based on "Karcher means" (Karcher, 2014, Zemel & Panaretos, 2019), which is more restrictive still. This implies that their method requires all involved measures to be absolutely continuous measures with densities that are bounded away from zero, with the target measure lying in the convex hull of the control measures.

## 1.2 Our contribution

In contrast to the existing approaches, our method is applicable for general probability measures, allows for the target measure to be outside the generalized geodesic convex hull of the control measures, can be implemented by a standard constrained linear regression, and provides a global—and in many cases unique—solution. The proposed method transforms the projection problem on the positively curved Wasserstein space into a linear optimization problem in the geometric tangent cone, which can be implemented via a linear regression. This problem takes the form of a deformable template (Boissard et al., 2015, Yuille, 1991), which connects our approach to this literature. Our method can be implemented in three steps: (i) obtain the general tangent cone structure at the target measure, (ii) construct a tangent space from the tangent cone via barycentric projections if it does not exist, and (iii) perform a linear regression to carry out the projection in the tangent space. This implementation of the projection approach via linear regression is computationally efficient, in particular compared to the existing methods in Bonneel et al. (2016) and Werenski et al. (2022).

The challenging part of the implementation is lifting the problem to the tangential structure: this requires computing the corresponding optimal transport plans between the target and each measure used in the projection. Many methods have been developed for this, see for instance Benamou & Brenier (2000), Jacobs & Léger (2020), Makkuva et al. (2020), Peyré & Cuturi (2019), Ruthotto et al. (2020) and references therein. Other alternatives compute approximations of the optimal transport plans via regularized optimal transport problems (Peyré & Cuturi, 2019), such as entropy regularized optimal transport (Galichon & Salanié, 2010, Cuturi, 2013). The proposed projection approach is compatible with any such method, therefore its complexity scales with that of estimating optimal transport plans. We provide results for the statistical consistency when estimating the measures via their empirical counterparts in practice.

To demonstrate the efficiency and utility of the proposed method, we apply our method in different settings and compare it to existing benchmarks such as Werenski et al. (2022). Furthermore, we extend the classical synthetic control estimator (Abadie & Gardeazabal, 2003, Abadie et al., 2010) to settings with observed individual heterogeneity in multivariate outcomes. The synthetic controls estimator is a projection approach, where one tries to predict an aggregate outcome of a treated unit by an optimal convex combination of control units and to use the weights of this optimal combination to construct the counterfactual state of the treated unit had it not received treatment. The novelty of our application is that it lets us perform the synthetic control method on the joint distribution of several outcomes, which complements the recently introduced method in Gunsilius (2022) designed for univariate outcomes. The possibility to project entire probability measures allows us to disentangle treatment heterogeneity at the treatment unit level. The possibility of working with general probability measures is key in this setting, as many outcomes of interest are not regular. We illustrate this by applying our method to estimate the effects of a Medicaid expansion policy in Montana, where we consider—as outcome—non-regular probability measure in $d = 28$ dimensions.

## 2 METHODOLOGY

### 2.1 THE 2-WASSERSTEIN SPACE $\mathcal{W}_2(\mathbb{R}^d)$

**The 2-Wasserstein Distance**    For probability measures $P_X, P_Y \in \mathscr{P}(\mathbb{R}^d)$ with supports $\mathcal{X}, \mathcal{Y} \subseteq \mathbb{R}^d$, respectively, the 2-Wasserstein distance $W_2(P_X, P_Y)$ is defined as

$$W_2(P_X, P_Y) \triangleq \left( \min_{\gamma \in \Gamma(P_X, P_Y)} \int_{\mathcal{X} \times \mathcal{Y}} |x - y|^2 \, \mathrm{d}\gamma(x, y) \right)^{\frac{1}{2}}. \tag{2.1}$$

Here, $|\cdot|$ denotes the Euclidean norm on $\mathbb{R}^d$ and

$$\Gamma(P_X, P_Y) \triangleq \left\{ \gamma \in \mathscr{P}(\mathbb{R}^d \times \mathbb{R}^d) : (\pi_1)_{\#}\gamma = P_X, \ (\pi_2)_{\#}\gamma = P_Y \right\}$$

is the set of all couplings of $P_X$ and $P_Y$. The maps $\pi_1$ and $\pi_2$ are the projections onto the first and second coordinate, respectively, and $T_{\#}P$ denotes the pushforward measure of $P$ via $T$, i.e. for any measurable $A \subseteq \mathcal{Y}$, $T_{\#}P(A) \equiv P(T^{-1}(A))$. An optimal coupling $\gamma \in \Gamma(P_X, P_Y)$ solving the optimal transport problem equation 2.1 is an *optimal transport plan*. By Prokhorov's theorem, a solution always exists in our setting. When $P_X$ is regular, i.e. when it does not give mass to sets of lower Hausdorff dimension in its support, then the optimal transport plan $\gamma$ solving equation 2.1 is unique and takes the form $\gamma = (\mathrm{Id} \times \nabla \varphi)_{\#} P_X$, where $\mathrm{Id}$ is the identity map on $\mathbb{R}^d$ and $\nabla \varphi(x)$ is the gradient of some convex function. This result is known as Brenier's theorem (Brenier, 1991, McCann, 1997, Villani, 2003, Theorem 2.12). By definition, all measures that possess a density with respect to Lebesgue measure are regular. Our main contribution is to allow for general probability measures, where only optimal transport plans but no maps exist.

**The 2-Wasserstein Space**    The 2-Wasserstein space $\mathcal{W}_2 \triangleq (\mathscr{P}_2(\mathbb{R}^d), W_2)$ is the metric space defined on the set $\mathscr{P}_2(\mathbb{R}^d)$ of all probability measures with finite second moments supported on $\mathbb{R}^d$, with the 2-Wasserstein distance as the metric. It is a geodesically complete space in the sense that between any two measures $P, P' \in \mathcal{W}_2$, one can define a geodesic $P_t : [0, 1] \to \mathcal{W}_2$ via the interpolation $P_t \triangleq (\pi_t)_{\#}\gamma$, where $\gamma$ is an optimal transport plan and $\pi_t : \mathbb{R}^d \times \mathbb{R}^d \to \mathbb{R}^d$ is defined through $\pi_t(x, y) \triangleq (1-t)x + ty$ (Ambrosio et al., 2008, McCann, 1997). Using this, it can be shown that $\mathcal{W}_2$ is a positively curved metric space $d > 1$ (Ambrosio et al., 2008, Theorem 7.3.2) and flat for $d = 1$ (Kloeckner, 2010), where curvature is defined in the sense of Aleksandrov (1951). This difference in the curvature properties is the main reason for why the multivariate setting requires different approaches compared to the established results for measures on the real line.

### 2.2 TANGENT CONE STRUCTURE ON $\mathcal{W}_2$

We exploit a tangential structure that can be defined for general measures on $\mathcal{W}_2$ (Ambrosio et al., 2008, Otto, 2001, Takatsu & Yokota, 2012). In particular, it allows us to circumvent solving a bilevel optimization problem as the one considered in Bonneel et al. (2016), whose statement we have included in the appendix.

The tangential structure relies on the fact that geodesics $P_t$ in $\mathcal{W}_2$ are linear in the transport plans $(\pi_t)_{\#}\gamma$. This implies a geometric tangent cone structure at each measure $P \in \mathcal{W}$ that can be defined as the closure in $\mathscr{P}_2(\mathbb{R}^d)$ of the set

$$\mathcal{G}(P) \triangleq \left\{ \gamma \in \mathscr{P}_2(\mathbb{R}^d \times \mathbb{R}^d) : (\pi_1)_{\#}\gamma = P, \ (\pi_1, \pi_1 + \varepsilon \pi_2)_{\#}\gamma \text{ is optimal for some } \varepsilon > 0 \right\}$$

with respect to the local distance

$$W_P^2(\gamma_{12}, \gamma_{13}) \triangleq \min \left\{ \int_{(\mathbb{R}^d)^3} |x_2 - x_3|^2 \, \mathrm{d}\gamma_{123} : \gamma_{123} \in \Gamma_1(\gamma_{12}, \gamma_{13}) \right\}, \tag{2.2}$$

where $\gamma_{12}$ and $\gamma_{13}$ are couplings between $P$ and some other measures $P_2$ and $P_3$, respectively, and $\Gamma_1(\gamma_{12}, \gamma_{13})$ is the set of all 3-couplings $\gamma_{123}$ such that the projection of $\gamma_{123}$ onto the first two elements is $\gamma_{12}$ and the projection onto the first and third element is $\gamma_{13}$ (Ambrosio et al., 2008,

Appendix 12). We can then define the exponential map at $P$ with respect to some tangent element $\gamma \in \mathcal{G}(P)$ by

$$\exp_P(\gamma) = (\pi_1 + \pi_2)_{\#}\gamma \, .$$

This tangent cone can be constructed at every $P \in \mathcal{W}$, irrespective of its support; in particular, we do not assume that the corresponding measures are regular, i.e., give mass to subsets of $\mathbb{R}^d$ of lower Hausdorff dimension. In the case where $P$ is regular the tangent cone structure reduces to a tangent space (Ambrosio et al., 2008, Theorem 8.5.1). This tangent space structure has been exploited in Mérigot et al. (2020) and Werenski et al. (2022), and we include the results for our projection approach in this special case in Appendix A.

## 2.3 Tangential Wasserstein projections

Our main contribution is to define a projection approach between general probability measures, where the target need not be regular. To define this notion of projection, we need to first define an appropriate notion of a geodesic convex hull. The novelty here is that we define this notion via generalized geodesics (Ambrosio et al., 2008, section 9.2) *centered at the target measure* $P_0$. For this, we extend the definition of $W_P$ to the multimarginal setting, by defining, for given couplings $\gamma_{0j} \in \Gamma(P_0, P_j), j \in [\![J]\!]$

$$W^2_{P_0;\lambda}(\gamma_{01}, \gamma_{02}, \ldots, \gamma_{0J}) \triangleq \min\left\{ \int_{(\mathbb{R}^d)^{J+1}} \sum_{j=1}^{J} \lambda_j \left|x_j - x_0\right|^2 d\boldsymbol{\gamma} : \boldsymbol{\gamma} \in \Gamma_1(\gamma_{01}, \ldots, \gamma_{0J}) \right\}, \tag{2.3}$$

where $\Gamma_1(\gamma_{01}, \ldots, \gamma_{0J}) \subseteq \Gamma(P_0, P_1, \ldots, P_J)$ is the set of all $(J+1)$-couplings $\boldsymbol{\gamma}$ such that the projection of $\boldsymbol{\gamma}$ onto the first- and $j$-th element is $\gamma_{0j}$. Note that this definition is similar to the multimarginal definition of the 2-Wasserstein barycenter (Agueh & Carlier, 2011, Gangbo & Święch, 1998), but "centered" at $P_0$. Based on this, we define the generalized geodesic convex hull of measures $\{P_j\}_{j \in [\![J]\!]}$ with respect to the measure $P_0$ as

$$\mathfrak{Co}_{P_0}\left(\{P_j\}_{j=1}^{J}\right) \triangleq \left\{ P(\lambda) \in \mathscr{P}_2(\mathbb{R}^d) : P(\lambda) = \left(\sum_{j=1}^{J} \lambda_j \pi_{j+1}\right)_{\#} \boldsymbol{\gamma}, \right.$$

$$\left. \boldsymbol{\gamma} \text{ solves } W^2_{P_0;\lambda}(\gamma_{01}, \ldots, \gamma_{0J}), \ \gamma_{0j} \text{ is optimal in } \Gamma(P_0, P_j) \ \forall j \in [\![J]\!], \quad \lambda \in \Delta^J \right\} . \tag{2.4}$$

A direct application of our tangential projection idea would lead us to solving

$$\lambda^* \triangleq \underset{\lambda \in \Delta^J}{\arg\min} \, W^2_{P_0;\lambda}(\gamma_{01}, \ldots, \gamma_{0J}) \, , \tag{2.5}$$

which would be a computationally prohibitive bilevel optimization problem similar to the one in Bonneel et al. (2016). We therefore rely on barycentric projections to reduce the general cone structure to a regular tangent space which we denote by $\mathcal{T}_{P_0}\mathcal{W}_2$ (Ambrosio et al., 2008). In this structure the projection problem equation 2.5 is replaced by

$$\lambda^* \triangleq \underset{\lambda \in \Delta^J}{\arg\min} \left\| \sum_{j=1}^{J} \lambda_j \left(b_{\gamma_{0j}} - \mathrm{Id}\right) \right\|^2_{L^2(P_0)} \, , \quad \text{with} \quad b_{\gamma_{0j}}(x_1) \triangleq \int_{\mathbb{R}^d} x_2 \, d\gamma_{0j, x_1}(x_2) \tag{2.6}$$

denoting the barycentric projections of optimal transport plans $\gamma_{0j}$ between $P_0$ and $P_j$. Here, $\gamma_{x_1}$ denotes the disintegration of the optimal transport plan $\gamma$ with respect to $P_0$.

This approach is a natural extension of the regular setting to general probability measures for two reasons. First, if the optimal transport plans $\gamma_{0j}$ are actually induced by some optimal transport map, then $b_{\gamma_{0j}}$ reduces to this optimal transport map; in this case the general tangent cone $\mathcal{G}(P_0)$ reduces to the regular tangent cone $\mathcal{T}_{P_0}\mathcal{W}_2$ (Ambrosio et al., 2008, Theorem 12.4.4). Second, by the

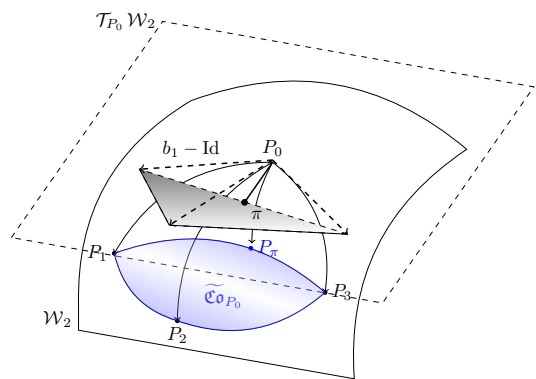

Figure 1: Tangential Wasserstein projection for a general target $P_0$.

$\mathcal{T}_{P_0}\mathcal{W}_2$ is the regular tangent space constructed by applying barycentric projection to $\mathcal{G}(P_0)$, the general tangent cone anchored at $P_0$. Thick dashed lines are tangent vectors $(b_j - \text{Id})$ generated by the respective barycentric projections. The gray shaded region is their convex hull in this constructed tangent space and $\pi$ is the projection of Id onto this convex hull. $P_\pi \triangleq \exp_{P_0}(\pi)$ is the projection of $P_0$ onto the generalized geodesic convex hull $\widetilde{\mathfrak{Co}}_{P_0}\left(\{P_1, P_2, P_3\}\right) \subseteq \mathcal{W}_2$ (blue).

definition of $b_\gamma$ and disintegrations in conjunction with Jensen's inequality it holds for all $\lambda \in \Delta^J$ that

$$\left\|\sum_{j=1}^{J} \lambda_j \left(b_{\gamma_{0j}} - \text{Id}\right)\right\|^2_{L^2(P_0)} \leqslant W^2_{P_0;\lambda}(\gamma_{01}, \ldots, \gamma_{0J}) . \tag{2.7}$$

This implies that for general $P_0$ we can also define a convex hull based on barycentric projections, which is of the form

$$\widetilde{\mathfrak{Co}}_{P_0}\left(\{P_j\}_{j=1}^{J}\right) \triangleq \left\{P(\lambda) \in \mathscr{P}_2(\mathbb{R}^d) : P(\lambda) = \left(\sum_{j=1}^{J} \lambda_j b_{\gamma_{0j}}\right)_\# P_0, \quad \lambda \in \Delta^J\right\} . \tag{2.8}$$

Furthermore, the contraction property equation 2.7 implies that $\mathfrak{Co}_{P_0} \subseteq \widetilde{\mathfrak{Co}}_{P_0}$, with equality when all transport plans are achieved via maps $\nabla\varphi_j$. Using these definitions, the following defines our notion of projection for general $P_0$ and shows that it projects onto $\widetilde{\mathfrak{Co}}_{P_0}$.

**Proposition 2.1.** *Consider a general target measure $P_0$ and a set $\{P_j\}_{j \in [\![J]\!]}$ of general control measures. Construct the measure $\widetilde{P}_\pi$ as*

$$\widetilde{P}_\pi \triangleq \exp_{P_0}\left(\sum_{j=1}^{J} \lambda_j^* b_{\gamma_{0j}} - \text{Id}\right) ,$$

*where the optimal weights $\lambda^* \in \Delta^J$ are obtained by solving equation 2.6 and $\gamma_{0j}$ are optimal plans transporting $P_0$ to $P_j$, respectively. Then for given optimal plans $\gamma_{0j}$, $\widetilde{P}_\pi$ is the unique metric projection of $P_0$ onto $\widetilde{\mathfrak{Co}}_{P_0}\left(\{P_j\}_{j=1}^{J}\right)$.*

The optimal plans $\gamma_{0j}$ transporting $P_0$ to $P_j$ need not be unique if $P_j$ lies outside the cut locus of $P_0$, i.e., when there is more than one optimal way to transport $P_0$ onto $P_j$. However, the projection for fixed $\gamma_{0j}$ is always unique by virtue of the linear regression.

## 3 STATISTICAL PROPERTIES OF THE WEIGHTS AND PROJECTION

We now provide statistical consistency results for our method when the corresponding measures $\{P_j\}_{j \in [\![J]\!]}$ are estimated from data. We consider the case where the measures $P_j$ are replaced by

their empirical counterparts

$$\mathbb{P}_{N_j}(A) \triangleq N_j^{-1} \sum_{n=1}^{N_j} \delta_{X_n}(A)$$

for every measurable set $A$ in the Borel $\sigma$-algebra on $\mathbb{R}^d$, where $\delta_x(A)$ is the Dirac measure and $(X_{1j}, \ldots, X_{N_j,j})$ is an independent and identically distributed set of random variables whose distribution is $P_j$. We explicitly allow for different sample sizes $\bigcup_{j=0}^{J} N_j = N$ for the different measures. To save on notation we write $\widehat{\varphi}_{N_j} \equiv \widehat{\varphi}_j$, $\widehat{b}_{0j} \equiv \widehat{b}_{\gamma_{0j}, N_j}$ and $\widehat{\gamma}_{0j} \equiv \widehat{\gamma}_{N_j, N_0}$ in the following.

**Proposition 3.1** (Consistency of the optimal weights). *Let $\left\{ \mathbb{P}_{N_j} \right\}_{j=0}^{J}$ be the empirical measures corresponding to the data $\left( X_{1j}, \ldots, X_{N_j j} \right)_{j=0}^{J}$ which are independent and identical draws from $P_j$, respectively, and are supported on some common latent probability space $(\Omega, \mathscr{A}, P)$. Assume all $P_j$ have finite second moments. As $N_j \to \infty$ for all $j \in [\![ J ]\!]$, the corresponding optimal weights $\widehat{\lambda}_N^* = \left( \widehat{\lambda}_{N_1}^*, \ldots, \widehat{\lambda}_{N_J}^* \right) \in \Delta^J$ obtained via*

$$\widehat{\lambda}_N^* \triangleq \arg\min_{\lambda \in \Delta^J} \left\| \sum_{j=1}^{J} \lambda_j \left( \widehat{b}_{0j} - \mathrm{Id} \right) \right\|_{L^2(\mathbb{P}_{N_0})}^2 , \qquad (3.1)$$

*satisfy*

$$P\left( \left| \widehat{\lambda}_N^* - \lambda^* \right| > \varepsilon \right) \to 0 \qquad \text{for all } \varepsilon > 0 \, ,$$

*where $\lambda^*$ solve equation 2.6.*

This consistency result directly implies consistency of the optimal weights in case the optimal transport problems between $\mathbb{P}_{N_0}$ and each $\mathbb{P}_{N_j}$ are achieved by optimal transport maps $\nabla \widehat{\varphi}_{N_j}$. We also have a consistency result for the empirical counterparts $\widetilde{\mathbb{P}}_{\pi, N}$ of the optimal projection $\widetilde{P}_\pi$.

**Corollary 3.1** (Consistency of the optimal projections). *In the setting of Proposition 3.1, the estimated projections $\widetilde{\mathbb{P}}_{\pi, N}$ converge weakly in probability to the projection $\widetilde{P}_\pi$ as $N_j \to \infty$ for all $j \in [\![ J ]\!]$.*

Proposition 3.1 and Corollary 3.1 hold in all generality and without any assumptions on the corresponding measures $P_j$, except that they possess finite second moments. To get stronger results, for instance parametric rates of convergences, one needs to make strong regularity assumptions on the measures $P_j$. Without these, the rate of convergence of optimal transport maps in terms of expected square loss is as slow as $n^{-2/d}$ (Hütter & Rigollet, 2021). Under such additional regularity conditions, the results for the asymptotic properties are standard, because the proposed method reduces to a classical semiparametric estimation problem, as the weights $\lambda_j$ are finite-dimensional.

## 4 ILLUSTRATIONS

### 4.1 MIXTURES OF GAUSSIANS

We consider mixtures of Gaussian in dimension $d = 10$. We draw from the following Gaussians:

$$\mathbf{X}_j \sim \mathcal{N}\left( \mu_j, \Sigma \right), \quad j = 0, 1, 2, 3 \, ,$$

where $\mu_0 = [10, 10, \ldots, 10]$, $\mu_1 = [50, 50, \ldots, 50]$, $\mu_2 = [200, 200, \ldots, 200]$, $\mu_3 = [-50, -50, \ldots, -50]$ and $\Sigma = \mathrm{Id}_{10} + 0.8 \, \mathrm{Id}_{10}^-$, with $\mathrm{Id}_{10}^-$ the $10 \times 10$ matrix with zeros on the main diagonal and ones on all off-diagonal terms. We then define the following mixtures: $\mathbf{Y}_0$ as target, and $\mathbf{Y}_1$, $\mathbf{Y}_2$, and $\mathbf{Y}_3$ as controls, where

$$\mathbf{Y}_0 = 0.7\mathbf{X}_0 + 0.15\mathbf{X}_1 + 0.15\mathbf{X}_2 \, , \qquad \mathbf{Y}_1 = 0.6\mathbf{X}_0 + 0.3\mathbf{X}_1 + 0.1\mathbf{X}_2 \, ,$$
$$\mathbf{Y}_2 = 0.7\mathbf{X}_1 + 0.2\mathbf{X}_2 + 0.1\mathbf{X}_3 \, , \qquad \mathbf{Y}_3 = 0.3\mathbf{X}_0 + 0.1\mathbf{X}_2 + 0.6\mathbf{X}_3 \, .$$

Each sample is 10000 points. The estimated optimal weights are $\lambda^* = [0.4329, 0.4002, 0.1669]$ with corresponding projection $\widetilde{\mathbf{Y}}_0$. With only 3 control units, it is not possible to perfectly replicate the entire target distribution. Still, in Figure 2, the optimal projection approximates $\mathbf{Y}_0$ reasonably well. Moreover, the weights are non-sparse in this case, indicating that the target $P_{Y_0}$ lies inside the geodesic convex hull of the control measures. In many real-world applications we observe sparse optimal weights; see, for instance, Section 4.3, and our application to synthetic controls in Section 5. This implies that in these settings the target lies outside the geodesic convex hull of the controls and is projected onto one of the faces.

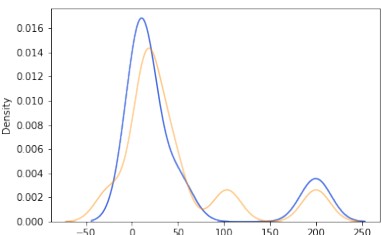

Figure 2: Kernel density estimates of the average of all dimensions comparing target $P_{\mathbf{Y}_0}$ (blue) and its projection $\widetilde{P}_{\pi}$ (orange) onto the generalized geodesic convex hull of $\{P_{\mathbf{Y}_1}, P_{\mathbf{Y}_2}, P_{\mathbf{Y}_3}\}$.

## 4.2    IMAGE EXPERIMENT: MNIST

We compare our results to those from the experiment in Section 4.3 of Werenski et al. (2022). We follow the experimental procedure described therein, taking as experimental data the MNIST dataset of $28 \times 28$ pixel images of hand-written digits (LeCun, 1998). We show comparison to the test case with image occlusion. We treat the normalized matrix as probability measures supported on a $28 \times 28$ grid. Figure 3 shows our results. We are able to more clearly replicate the edges and contours of the target image, compared to both the Euclidean projection and the method described in Werenski et al. (2022). Moreover, our method manages to replicate the overall shape of the specific handwritten number closer than the other methods; in particular, it is the only method that correctly replicates the horizontal bar at the bottom of this particular handwritten "4".

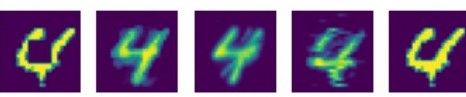

Figure 3: Left to right: occluded image; Euclidean projection; result from Werenski et al. (2022), using optimal weights obtained from their method; result from our approach, using optimal weights obtained from equation 2.6; target image.

## 4.3    IMAGE EXPERIMENT: LEGO BRICKS

To examine the general properties of how our method obtains the optimal weights, we provide an application on replicating a target image of an object using images of the same object taken from different angles. We use the Lego Bricks dataset available from `Kaggle`, which contains approximately 12,700 images of 16 different Lego bricks in RGBA format. Our method manages to replicate the target block rather well, while only using the information of control units that look sufficiently like the target. In particular, in replication, our method does not use information from any image of the underside of the Lego brick. In contrast, the Euclidean projection does not provide the correct rotation in the replication, and suffers from the standard blur induced by using a mixture of images.

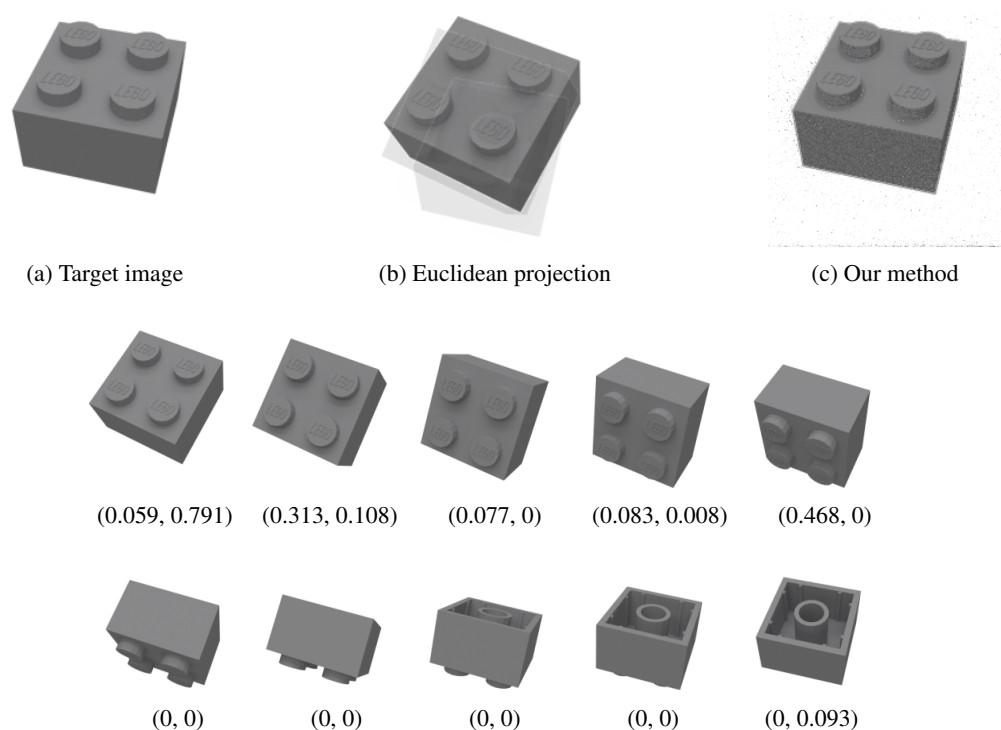

(a) Target image     (b) Euclidean projection     (c) Our method

(0.059, 0.791)   (0.313, 0.108)   (0.077, 0)   (0.083, 0.008)   (0.468, 0)

(0, 0)   (0, 0)   (0, 0)   (0, 0)   (0, 0.093)

Figure 4: Top row: target block, Euclidean projection, and projection from our method. Middle and bottom rows: Control units used in simulation. Left entry in parentheses is optimal weights from our method, right entry are optimal weights from Euclidean projection. Weights are denoted as zero if they are less than 1e-6.

## 5 APPLICATION TO CAUSAL INFERENCE VIA SYNTHETIC CONTROLS

When analyzing the causal effect of treatment on a unit, such as that of public policies or medical interventions, there is often no comparable control unit that can capture the treated unit's underlying characteristics. The classical synthetic controls method (Abadie & Gardeazabal, 2003, Abadie et al., 2010) aims to create a suitable control unit by replicating the pre-treatment outcome trends of the treated unit, using some optimally chosen set of control units. This is achieved by projecting the observed characteristics of the target unit onto the convex hull defined by the characteristics of control units in the pre-treatment periods. The optimal weights obtained by this projection, therefore, describe how much each control unit contributes to the target unit's counterfactual outcome in the post-treatment period (Abadie, 2021).

We apply our notion of projections to extend the classical synthetic control method to work on joint measures of several outcomes, which allows to disentangle heterogeneous treatment effects and complements the univariate method introduced in Gunsilius (2022). As demonstration, we study the effect of health insurance coverage following state-level Medicaid expansion in Montana in 2016. The variables of interest are Medicaid coverage, employment status, log wages, and log hours worked. For control units, we use the twelve states for which such expansion has never occurred; these are: Alabama, Florida, Georgia, Kansas, Mississippi, North Carolina, South Carolina, South Dakota, Tennessee, Texas, Wisconsin, Wyoming. Additional information can be found in Appendix C.

We estimate "synthetic Montana", i.e. Montana had it not adopted Medicaid expansion, by estimating the optimal weights $\lambda^*$ using data from 2010 to 2016, and solving equation 2.6 over the joint distribution of the four outcomes over the time period from 2010 to 2016, which generates measures in $d = 28$ dimensions. We note that we estimate one set of optimal weights—specifically, one for each control state—over the entire time period. We then estimate the counterfactual joint distribution using data from 2017 to 2020, by using the optimal weights $\lambda^*$ and computing the weighted

barycenter (Agueh & Carlier, 2011) of the control states using these weights. Details of sample selection and estimating "synthetic Montana" are described in Appendix C. The results of the general causal effect of the Medicaid expansion policy in Montana averaged over the years $2017 - 2020$ are illustrated in Figure 5.

Consistent with findings in Courtemanche et al. (2017), Mazurenko et al. (2018), we find significant first- and second order effects of Medicaid expansion, which are summarized in the top row and the bottom row of Figure 5, respectively. "Synthetic Montana" has much lower proportion of individuals insured under Medicaid, suggesting that expanding Medicaid eligibility directly affects the extensive margin of Medicaid enrollment. The disemployment effect is less pronounced in comparison to the enrollment effect we estimated, but nonetheless positive and nontrivial, consistent with the findings in, e.g., Peng et al. (2020), but inconsistent with those in, e.g., Gooptu et al. (2016). We also find positive second-order effects, summarized in the bottom row of Figure 5. Additional details are in Appendix C.

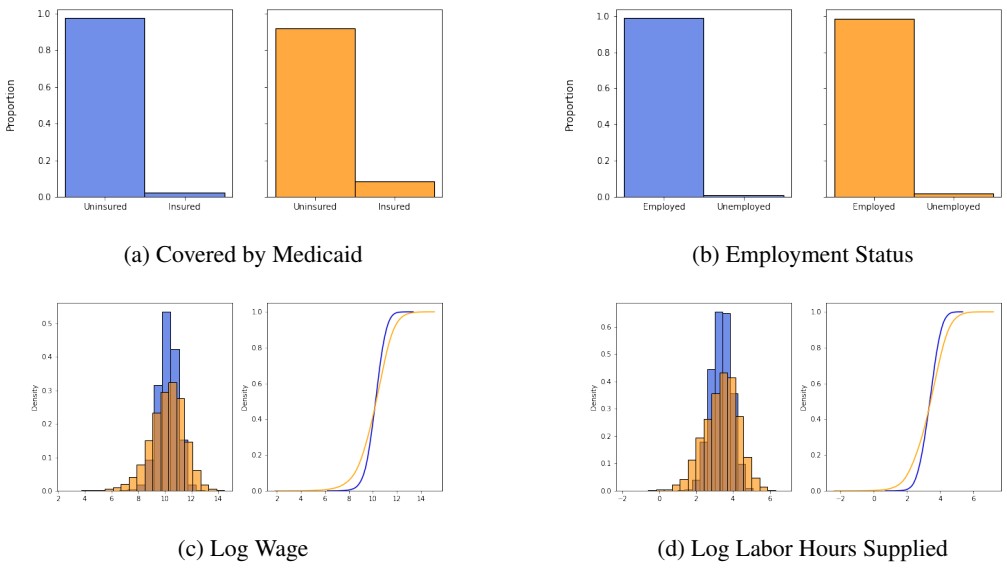

(a) Covered by Medicaid

(b) Employment Status

(c) Log Wage

(d) Log Labor Hours Supplied

Figure 5: Counterfactual (blue) vs actual (orange) Montana from 2017 to 2020. In the bottom row, histograms of data distributions are shown on the left, and cumulative distribution functions are shown on the right.

## 6    CONCLUSION

We have developed a projection method between sets of probability measures supported on $\mathbb{R}^d$ based on the tangent cone structure of the 2-Wasserstein space. Our method seeks to best approximate some general target measure using some chosen set of control measures. In particular, it provides a global (and in most cases unique) optimal solution. Our application to evaluating the first- and second-order effects of Medicaid expansion in Montana via an extension of the synthetic controls estimator (Abadie & Gardeazabal, 2003, Abadie et al., 2010) demonstrates the method's utility in allowing for a method that is applicable for general probability measures. The method still works without restricting optimal weights to be in the unit simplex, which would allow for extrapolation beyond the convex hull of the control units, providing a notion of tangential regression. It can also be extended to a continuum of measures, using established consistency results of barycenters (e.g. Le Gouic & Loubes, 2017).

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

# APPENDIX

## A WASSERSTEIN BARYCENTERS AND THE SPECIAL CASE OF A REGULAR TARGET MEASURE

The natural approach to defining projections on $\mathcal{W}_2$ is to work on the manifold directly. As mentioned in the main text, this leads to a bilevel optimization problem, based on the notion of barycenters in Wasserstein space (Agueh & Carlier, 2011, Carlier & Ekeland, 2010):

$$\bar{P}(\lambda) = \underset{P \in \mathscr{P}_2(\mathbb{R}^d)}{\arg\min} \sum_{j=1}^{J} \frac{\lambda_j}{2} W_2^2(P, P_j).$$

With this definition, and assuming that the barycenter $\bar{P}(\lambda)$ is unique for given $\lambda$, the bilevel projection problem reads:

$$\lambda^* \in \underset{\lambda \in \Delta^J}{\arg\min} \ W_2(P_0, \bar{P}(\lambda)), \qquad \text{where} \quad \bar{P}(\lambda) = \underset{P \in \mathscr{P}_2(\mathbb{R}^d)}{\arg\min} \sum_{j=1}^{J} \frac{\lambda_j}{2} W_2^2(P, P_j). \qquad \text{(A.1)}$$

A version of this approach is used in Bonneel et al. (2016) to define a notion of regression between probability measures in low dimensions. The challenges here are mathematical and computational. Importantly, the optimal weights $\lambda^*$ need not be unique. This is not an issue for the applications considered in Bonneel et al. (2016), like color transport; however, it is important in statistical settings when the weights convey information used in further procedures, like causal inference via synthetic controls, where the optimal weights are used to introduced a counterfactual outcome of a treated unit had it not been treated (Abadie & Gardeazabal, 2003, Abadie et al., 2010, Abadie, 2021). Moreover, the bi-level optimization structure makes solving the problem prohibitively costly in higher dimensions. Bonneel et al. (2016) introduce a gradient descent approach based on an entropy-regularized analogue of $W_2$ (Cuturi, 2013, Peyré & Cuturi, 2019) that can be implemented in low-dimensional settings.

Other approaches like Werenski et al. (2022) introduce a tangential approach, but under strong assumptions on the involved measures: they need to be absolutely continuous with densities bounded away from zero on their support, and in particular the target measure must be known to lie inside the convex hull of the other measures. A starting point for this is to consider a characterization of the barycenter $\bar{P}(\lambda)$ for fixed weights of a set $\{P_j\}_{j \in \llbracket J \rrbracket}$ in regular tangent spaces. Agueh & Carlier (2011, Equation (3.10)) show that if at least one of the measures is absolutely continuous with respect to Lebesgue measure, then $\bar{P}(\lambda)$ can be characterized via

$$\sum_{j=1}^{J} \lambda_j \left( \nabla \tilde{\varphi}_j - \text{Id} \right) = 0, \qquad \text{(A.2)}$$

where $\{\tilde{\varphi}_j\}_{j \in \llbracket J \rrbracket}$ are the optimal transport maps from the barycenter to the respective measure $P_j$, i.e. $(\tilde{\varphi}_j)_{\#} \bar{P}(\lambda) = P_j$. Each term of the summand in equation A.2 is an element in $\mathcal{T}_{\bar{P}(\lambda)} \mathcal{W}_2(\mathbb{R}^d)$ by construction.

More generally, the condition equation A.2 is a sufficient condition for $\bar{P}(\lambda)$ to be a "Karcher mean" (Karcher, 2014) in $\mathcal{W}_2$ (Zemel & Panaretos, 2019). In fact, a "Karcher mean" of a set of measures $\{P_j\}_{j \in \llbracket J \rrbracket}$ is defined as the gradient of the Fréchet functional in $\mathcal{W}_2$ and is characterized through equation A.2 holding $\bar{P}(\lambda)$-almost everywhere. equation A.2 is a stronger condition because it is assumed to hold at every point in the support of $\bar{P}(\lambda)$, not just almost every point. Álvarez-Esteban et al. (2016) use this characterization to introduce a fixed-point approach to compute Wasserstein barycenters, and Werenski et al. (2022) use this structure to introduce a replication approach for absolutely continuous measures whose densities are bounded away from zero and whose target measure lies inside the convex hull of the control measures. Related is the recent definition of weak barycenters in Cazelles et al. (2021), where the authors replace the optimal transport maps from the classical optimal transport problem by the weak optimal transport problem introduced in Gozlan et al. (2017). Heuristically, this characterization is that of a *deformable template*. A measure $P$ is a deformable template if there exists a set of deformations $\{\psi_j\}_{j=1,\dots,J}$ such that $\psi_{j\#} P = P_j$, in a

way that their weighted average is "as close to the identity" as possible. In our setting $\psi_j \equiv \nabla\varphi_j - \mathrm{Id}$ (Anderes et al., 2015, Boissard et al., 2015, Yuille, 1991).

In our setting of interest, our tangential projection reduces to

$$\lambda^* \triangleq \underset{\lambda \in \Delta^J}{\arg\min} \left\| \sum_{j=1}^{J} \lambda_j \left( \nabla\varphi_j - \mathrm{Id} \right) \right\|_{L^2(P_0)}^2, \tag{A.3}$$

where $\nabla\varphi_j$ are the optimal transport maps between the target $P_0$ and the control measures $P_j$, $j \in [\![J]\!]$. In contrast to Werenski et al. (2022) the target measure does not need to lie inside the convex hull of the other measures.

Based on these definitions we can show that our approach is a projection of the target $P_0$ onto $\mathfrak{Co}_{P_0}\left(\{P_j\}_{j=1}^J\right)$ in the case where $P_0$ is regular.

**Proposition A.1.** *Consider a regular target measure $P_0$ and a set $\{P_j\}_{j\in[\![J]\!]}$ of general control measures. Construct the measure $P_\pi$ as*

$$P_\pi \triangleq \exp_{P_0}\left( \sum_{j=1}^{J} \lambda_j^*(\nabla\varphi_j - \mathrm{Id}) \right),$$

*where the optimal weights $\lambda^* \in \Delta^J$ are obtained by solving equation A.3 and $\nabla\varphi_j$ are the optimal maps transporting $P_0$ to $P_j$, respectively. Then $P_\pi$ is the unique metric projection of $P_0$ onto $\mathfrak{Co}_{P_0}\left(\{P_j\}_{j=1}^J\right)$.*

## B  PROOFS

*Proof of Proposition A.1.* Define the following closed and convex subset $\mathcal{C} \subseteq L^2(P_0)$ for fixed optimal transportation maps between $P_0$ and $P_j$, denoted $\nabla\varphi_j$:

$$\mathcal{C} \triangleq \left\{ f \in L^2(P_0) : f = \sum_{j=1}^{J} \lambda_j \nabla\varphi_j \text{ for some } \lambda \in \Delta^J \right\}.$$

Recall that the transport maps $\nabla\varphi_j$ exist since $P_0$ is regular. Using $\mathcal{C}$, we can rewrite equation A.3 as

$$\underset{\lambda \in \Delta^J}{\arg\min} \left\| \sum_{j=1}^{J} \lambda_j \nabla\varphi_j - \mathrm{Id} \right\|_{L^2(P_0)}^2 = \underset{f \in \mathcal{C}}{\arg\min} \| f - \mathrm{Id} \|_{L^2(P_0)}^2,$$

which by definition is the metric projection of $\mathrm{Id}$ onto $\mathcal{C}$. Since $\mathcal{C}$ is a non-empty closed and convex subset of the Hilbert space $L^2(P_0)$, this metric projection exists and is unique (Aliprantis & Border, 1999, Theorem 6.53). Moreover, if $\mathrm{Id} \in \mathcal{C}$, then $\pi_\mathcal{C} = \mathrm{Id}$; otherwise, $\pi_\mathcal{C} \in \partial\mathcal{C}$, where $\partial\mathcal{C}$ is the boundary of $\mathcal{C}$ (Aliprantis & Border, 1999, Lemma 6.54).

Since $P_0$ is regular, the exponential map is continuous. In fact, for every $j \neq k$,

$$W_2^2(P_j, P_k) = W_2^2((\nabla\varphi_j)_\# P_0, (\nabla\varphi_k)_\# P_0) \leqslant \int_{\mathbb{R}^d} \left| \nabla\varphi_j - \nabla\varphi_k \right|^2 \mathrm{d}P_0(x).$$

In other words, the distance between $P_j$ and $P_k$ in $\mathcal{W}_2(\mathbb{R}^d)$ is smaller than that between corresponding elements $\nabla\varphi_j, \nabla\varphi_k$ in the tangent space. This implies continuity of the exponential map.

Furthermore, in this regular setting, the exponential map sends convex sets in $\mathcal{T}_{P_0}\mathcal{W}_2$ to generalized geodesically convex sets in $\mathcal{W}_2$. Mechanically, for any two (scaled) elements $t(\nabla\varphi_j - \mathrm{Id})$ and $s(\nabla\varphi_k - \mathrm{Id})$ in $\mathcal{T}_{P_0}\mathcal{W}_2$, and any $\rho \in [0, 1]$,

$$\exp_{P_0}(\rho t(\nabla\varphi_j - \mathrm{Id}) + (1-\rho)s(\nabla\varphi_k - \mathrm{Id}))$$
$$= \exp_{P_0}((\rho t \nabla\varphi_j + (1-\rho)s \nabla\varphi_k) - (\rho t + (1-\rho)s)\,\mathrm{Id})$$

$$= \exp_{P_0} \left( \widetilde{\rho} \left[ \left[ \frac{\rho t}{\widetilde{\rho}} \nabla \varphi_j + \frac{(1-\rho)s}{\widetilde{\rho}} \nabla \varphi_k \right] - \mathrm{Id} \right] \right)$$

$$= \left( \left[ \rho t \nabla \varphi_j + (1-\rho)s \nabla \varphi_k \right] + (1-\widetilde{\rho})\,\mathrm{Id} \right)_{\#} P_0$$

$$= \left( \left[ \rho t (\nabla \varphi_j - \mathrm{Id}) + (1-\rho)s(\nabla \varphi_k - \mathrm{Id}) \right] + \mathrm{Id} \right)_{\#} P_0$$

where $\widetilde{\rho} \triangleq \rho t + (1-\rho)s$. This is a generalized geodesic connecting $P_j$ and $P_k$, via the optimal transport map between them and $P_0$ (Ambrosio et al., 2008, section 9.2). The same argument holds when extending generalized geodesics to generalized barycenters by taking convex combination of more measures than a binary interpolation with respect to $\rho$. Mechanically, for any $\lambda \in \Delta^J$ and $t_j > 0$ for all $j \in [\![J]\!]$,

$$\exp_{P_0} \left( \sum_{j=1}^{J} \lambda_j t_j (\nabla \varphi_j - \mathrm{Id}) \right) = \exp_{P_0} \left( \sum_{j=1}^{J} \lambda_j t_j \nabla \varphi_j - \sum_{j=1}^{J} \lambda_j t_j \,\mathrm{Id} \right)$$

$$= \exp_{P_0} \left( \widetilde{\rho}_J \left[ \sum_{j=1}^{J} \widetilde{\rho}_J \nabla \varphi_j - \mathrm{Id} \right] \right)$$

$$= \left( \left[ \sum_{j=1}^{J} \lambda_j t_j \nabla \varphi_j \right] + (1-\widetilde{\rho}_J)\,\mathrm{Id} \right)_{\#} P_0$$

$$= \left( \left[ \sum_{j=1}^{J} \lambda_j t_j (\nabla \varphi_j - \mathrm{Id}) \right] + \mathrm{Id} \right)_{\#} P_0$$

where $\widetilde{\rho}_J \triangleq \sum_{j=1}^{J} \lambda_j t_j$. This proves the exponential map is generalized geodesically convex.

From above it follows that $P_\pi \triangleq \exp_{P_0}(\pi_C)$ is either in the interior of $\mathcal{C}$, which is the case if $\mathrm{Id} \in \mathcal{C}$, or on its boundary: since $\pi_C \in \partial\mathcal{C}$, $\exp_{P_0}(\pi_C) \in \exp_{P_0}(\partial\mathcal{C})$. By continuity of the exponential map it follows that $\exp_{P_0}(\partial\mathcal{C}) = \partial \exp_{P_0}(\mathcal{C})$. Combining all steps above show that $P_\pi$ is a *geodesic* metric projection of $P_0$ onto the geodesic convex hull of $\{P_j\}_{j=1}^{J}$. ∎

*Proof of Proposition 2.1.* The result follows from the same argument as the proof of Proposition A.1. Theorem 12.4.4 in Ambrosio et al. (2008) shows that $\mathcal{T}_{P_0} \mathcal{W}_2$ is the image of the barycentric projection of measures in the general tangent cone: $b_\gamma(x)$ is an optimal transport map if $\gamma$ is an optimal transport plan. But the exponential map satisfies

$$\exp_{P_0}(v) = (v + \mathrm{Id})_{\#} P_0 \qquad \text{for all } v \in \mathcal{T}_{P_0} \mathcal{W}_2.$$

This implies that

$$\widetilde{P}_\pi \triangleq \exp_{P_0} \left( \sum_{j=1}^{J} \lambda_j^* b_{\gamma_{0j}} - \mathrm{Id} \right) = \left( \sum_{j=1}^{J} \lambda_j^* b_{\gamma_{0j}} \right)_{\#} P_0 \in \widetilde{\mathfrak{Co}}_{P_0} \left( \{P_j\}_{j=1}^{J} \right),$$

since the convex combination of elements in the subgradients of convex functions lie in the subgradient of a convex function (provided the subgradient of each convex function is nonempty, which is the case here). Then the continuity and generalized convexity of the exponential map for elements in the regular tangent space $\mathcal{T}_{P_0} \mathcal{W}_2$ implies the result. ∎

*Proof of Proposition 3.1.* We split the proof into two parts. In the first part we prove the convergence in probability of the family of objective functions equation 3.1 to their population counterparts equation 2.6 if the empirical measures $\mathbb{P}_{N_j}$ converge weakly in probability to the population measures $P_j$. In the second step we use the fact that $\widehat{\lambda}^*$ is a classical semiparametric estimator (Andrews, 1994, Newey & McFadden, 1994) to derive the convergence of the weights.

**Step 1: Convergence of the objective functions**   To show the convergence of the of the objective functions for obtaining the weights $\lambda^*$, we write

$$
\left| \left\| \sum_{j=1}^{J} \lambda_j b_{0j} - \mathrm{Id} \right\|_{L^2(P_0)}^2 - \left\| \sum_{j=1}^{J} \lambda_j \widehat{b}_{0j} - \mathrm{Id} \right\|_{L^2(\mathbb{P}_{N_0})}^2 \right|
$$

$$
= \left| \int \left| \sum_{j=1}^{J} \lambda_j b_{0j}(x) - x \right|^2 \mathrm{d}P_0 - \int \left| \sum_{j=1}^{J} \lambda_j \widehat{b}_{0j}(x) - x \right|^2 \mathrm{d}\mathbb{P}_{N_0} \right| .
$$

We hence want to show that

$$
\lim_{\bigwedge_j N_j \to \infty} \left| \int \left| \sum_{j=1}^{J} \lambda_j b_{0j}(x) - x \right|^2 \mathrm{d}P_0(x) - \int \left| \sum_{j=1}^{J} \lambda_j \widehat{b}_{0j}(x) - x \right|^2 \mathrm{d}\mathbb{P}_{N_0}(x) \right| = 0 ,
$$

where $\bigwedge_j N_j \equiv \min \{ N_0, \ldots, N_J \}$.

We split the result into two parts. The first part shows that

$$
\liminf_{\bigwedge_j N_j \to \infty} \int_{\mathbb{R}^d} \left| \sum_{j=1}^{J} \lambda_j \widehat{b}_{0j}(x_0) - x_0 \right|^2 \mathrm{d}\mathbb{P}_{N_0}(x_0) \geqslant \int_{\mathbb{R}^d} \left| \sum_{j=1}^{J} \lambda_j b_{0j}(x_0) - x_0 \right|^2 \mathrm{d}P_0(x_0).
$$

In the second part we use the $L^2(P_0)$ convergence of the barycentric projections to prove that the limit exists and coincides with the limit inferior.

For the first part, we have

$$
\liminf_{\bigwedge_j N_j \to \infty} \int_{\mathbb{R}^d} \left| \sum_{j=1}^{J} \lambda_j \widehat{b}_{0j}(x_0) - x_0 \right|^2 \mathrm{d}\mathbb{P}_{N_0}(x_0) =
$$

$$
\liminf_{\bigwedge_j N_j \to \infty} \int_{(\mathbb{R}^d)^{J+1}} \left| \sum_{j=1}^{J} \lambda_j x_j - x_0 \right|^2 \mathrm{d}\widehat{\gamma}_N(x_0, x_1, \ldots, x_J),
$$

where $\widehat{\gamma}_N(x_0, x_1, \ldots, x_J)$ is a measure that solves

$$
\min \left\{ \int_{(\mathbb{R}^d)^{J+1}} \sum_{j=1}^{J} \lambda_j \left| x_j - x_0 \right|^2 \mathrm{d}\gamma : \gamma \in \Gamma_1(\widehat{\gamma}_{01}, \ldots, \widehat{\gamma}_{0J}) \right\} ,
$$

$\widehat{\gamma}_{0j}$ are the optimal couplings between $\mathbb{P}_{N_0}$ and $\widetilde{\mathbb{P}}_{N_j} \triangleq \left( \widehat{b}_{0j} \right)_{\#} \mathbb{P}_{N_0}$. Since all measures are defined on the complete and separable space $\mathbb{R}^d$, and by assumption of finite second moments, i.e.

$$
\max_{j \in [\![J]\!]} \sup_{N_j} \int \left| x_j - x_0 \right|^2 \mathrm{d}\widehat{\gamma}_{0j} < +\infty ,
$$

it holds that each sequence $\widehat{\gamma}_{0j}$ is tight by Ulam's theorem (Dudley, 2018, Theorem 7.1.4). Using the fact that $\lambda \in \Delta^J$ and $\widehat{\gamma}_N \in \Gamma_1(\widehat{\gamma}_{01}, \ldots, \widehat{\gamma}_{0J})$, applying Jensen's inequality gives us

$$
\max_{j \in [\![J]\!]} \sup_{N_j} \int_{(R^d)^{J+1}} \left| \sum_{j=1}^{J} \lambda_j x_j - x_0 \right|^2 \mathrm{d}\widehat{\gamma}_N \leqslant \max_{j \in [\![J]\!]} \sup_{N_j} \sum_{j=1}^{J} \lambda_j \int_{\mathbb{R}^d} \left| x_j - x_0 \right|^2 \mathrm{d}\widehat{\gamma}_{0j} < +\infty ,
$$

which implies that $\widehat{\gamma}_N$ is tight. By Prokhorov's theorem, there exists a subsequence $\widehat{\gamma}_{N_k}$ that weakly converges to a limit measure $\gamma$. Therefore, by the continuity of the map $(x_0, x_j) \mapsto \sum_j \lambda_j x_j - x_0$, it follows from classical convergence results (Ambrosio et al., 2008, Lemma 5.1.12(d)) that

$$\liminf_{\bigwedge_j N_j \to \infty} \int_{(\mathbb{R}^d)^{J+1}} \left| \sum_{j=1}^J \lambda_j x_j - x_0 \right|^2 \mathrm{d}\widehat{\gamma}_N(x_0, x_1, \dots, x_J) =$$

$$\int_{(\mathbb{R}^d)^{J+1}} \left| \sum_{j=1}^J \lambda_j x_j - x_0 \right|^2 \mathrm{d}\boldsymbol{\gamma}(x_0, \dots, x_J).$$

Furthermore, by the same argument via Jensen's inequality, i.e.,

$$\int_{(\mathbb{R}^d)^{J+1}} \left| \sum_{j=1}^J \lambda_j x_j - x_0 \right|^2 \mathrm{d}\boldsymbol{\gamma}(x_0, \dots, x_J) \leqslant \sum_{j=1}^J \int_{(\mathbb{R}^d)^2} \left| \lambda_j x_j - x_0 \right|^2 \mathrm{d}\gamma_{0j}(x_0, x_j) < +\infty \,,$$

it follows that the limit $\boldsymbol{\gamma} \in \Gamma_1(\gamma_{01}, \dots, \gamma_{0J})$.

Now note that by the definition of disintegration it follows that (Ambrosio et al., 2008, Lemma 5.3.2)

$$\boldsymbol{\gamma} \in \Gamma_1(\gamma_{01}, \dots, \gamma_{0J}) \qquad \Longleftrightarrow \qquad \boldsymbol{\gamma}_{x_0} \in \Gamma\left(\gamma_{1|x_0}, \dots, \gamma_{J|x_0}\right) \,,$$

where

$$\boldsymbol{\gamma} = \int \boldsymbol{\gamma}_{x_0} \,\mathrm{d}P_0(x_0) \qquad \text{and} \qquad \gamma_{0j} = \int \gamma_{j|x_0} \,\mathrm{d}P_0(x_0)$$

are the disintegrations of $\boldsymbol{\gamma}$ and $\gamma_{0j}$ with respect to $P_0$, respectively. Therefore, we have

$$\int_{(\mathbb{R}^d)^{J+1}} \left| \sum_{j=1}^J \lambda_j x_j - x_0 \right|^2 \mathrm{d}\boldsymbol{\gamma}(x_0, \dots, x_J)$$

$$= \int_{\mathbb{R}^d} \int_{(\mathbb{R}^d)^J} \left| \sum_{j=1}^J \lambda_j x_j - x_0 \right|^2 \mathrm{d}\boldsymbol{\gamma}_{x_0}(x_1, \dots, x_J) \,\mathrm{d}P_0(x_0)$$

$$\geqslant \int_{\mathbb{R}^d} \left| \int_{(\mathbb{R}^d)^J} \left( \sum_{j=1}^J \lambda_j x_j - x_0 \right) \mathrm{d}\boldsymbol{\gamma}_{x_0}(x_1, \dots, x_J) \right|^2 \mathrm{d}P_0(x_0)$$

$$= \int_{\mathbb{R}^d} \left| \sum_{j=1}^J \lambda_j \int_{(\mathbb{R}^d)^J} x_j \,\mathrm{d}\boldsymbol{\gamma}_{x_0}(x_1, \dots, x_J) - x_0 \right|^2 \mathrm{d}P_0(x_0)$$

$$= \int_{\mathbb{R}^d} \left| \sum_{j=1}^J \lambda_j \int_{\mathbb{R}^d} x_j \,\mathrm{d}\gamma_{j|x_0}(x_j) - x_0 \right|^2 \mathrm{d}P_0(x_0)$$

$$= \int_{\mathbb{R}^d} \left| \sum_{j=1}^J \lambda_j b_{0j}(x_0) - x_0 \right|^2 \mathrm{d}P_0(x_0),$$

where the third lines follows from Jensen's inequality and the fifth line from $\boldsymbol{\gamma}_{x_0} \in \Gamma\left(\gamma_{1|x_0}, \dots, \gamma_{J|x_0}\right)$. This shows the first part.

For the second part we use the fact that each barycentric projection $\widehat{b}_{0j}(x_1)$ is an optimal transport map between $\mathbb{P}_{N_0}$ and $\widetilde{\mathbb{P}}_{N_j}$ if $\widehat{\gamma}_{0j}$ is an optimal transport plan between $\mathbb{P}_{N_0}$ and $\mathbb{P}_{N_j}$, which follows from Theorem 12.4.4 in Ambrosio et al. (2008). As before, we know that $\left(\widehat{b}_{0j}\right)_{\#} \mathbb{P}_{N_0}$ is a tight sequence that converges to some $\widetilde{P}_j$. By definition and the fact that $\hat{b}_{0j}$ is the gradient of a convex function between $\mathbb{P}_{N_0}$ and $\widetilde{\mathbb{P}}_{N_j}$, $\widehat{b}_{0j}$ is the unique optimal transport map between $\mathbb{P}_{N_0}$ and $\widetilde{\mathbb{P}}_{N_j}$ for all $N_j$ and all $j$. Since the measures $P_j$ have finite second moments by assumption, we have

$$\limsup_{N_0 \wedge N_j \to \infty} \int_{\mathbb{R}^d} |x_j|^2 \,\mathrm{d}\widetilde{\mathbb{P}}_{N_j} = \limsup_{N_0 \wedge N_j \to \infty} \int_{\mathbb{R}^d} \left| \widehat{b}_{0j}(x_0) \right|^2 \mathrm{d}\mathbb{P}_{N_0}$$

$$= \limsup_{N_0 \wedge N_j \to \infty} \int_{\mathbb{R}^d} \left| \int_{\mathbb{R}^d} x_j \, \mathrm{d}\widehat{\gamma}_{j|x_0}(x_j) \right|^2 \mathrm{d}\mathbb{P}_{N_0}$$

$$\leqslant \limsup_{N_0 \wedge N_j \to \infty} \int_{(\mathbb{R}^d)^2} \left| x_j \right|^2 \mathrm{d}\widehat{\gamma}_{0j}(x_0, x_j)$$

$$= \int_{(\mathbb{R}^d)^2} \left| x_j \right|^2 \mathrm{d}\gamma_{0j}(x_0, x_j) < +\infty,$$

where the last equality follows from the tightness of $\widehat{\gamma}_{0j}$, as shown earlier. Therefore, by standard stability results for optimal transport maps (Segers, 2022, Panaretos & Zemel, 2020), it holds that $\widehat{b}_{0j}$ converges uniformly on every compact subset $K \subseteq \mathbb{R}^d$ in the support of the limit measure $\widetilde{P}_j$, that is

$$\lim_{N_0 \wedge N_j \to \infty} \sup_{x_0 \in K} \left| \widehat{b}_{0j}(x_0) - v_j(x_0) \right| = 0 \,,$$

where $v_j$ is the optimal transport map between $P_0$ and $\widetilde{P}_j$.

We now show that $v_j = b_{0j}$ $P_0$-almost everywhere. From the local uniform convergence, we can then derive "strong $L^2$-convergence" (Ambrosio et al., 2008, Definition 5.4.3) of the potentials:

$$\limsup_{N_0 \wedge N_j \to \infty} \left| \left\| \widehat{b}_{0j} \right\|_{L^2(\mathbb{P}_{N_0})} - \left\| v_j \right\|_{L^2(P_0)} \right|$$

$$\leqslant \limsup_{N_0 \wedge N_j \to \infty} \left| \left\| \widehat{b}_{0j} \right\|_{L^2(\mathbb{P}_{N_0})} - \left\| v_j \right\|_{L^2(\mathbb{P}_{N_0})} \right| + \limsup_{N_0 \to \infty} \left| \left\| v_j \right\|_{L^2(\mathbb{P}_{N_0})} - \left\| v_j \right\|_{L^2(P_0)} \right|$$

$$\leqslant \limsup_{N_0 \wedge N_j \to \infty} \left\| \widehat{b}_{0j} - v_j \right\|_{L^2(\mathbb{P}_{N_0})} + \limsup_{N_0 \to \infty} \left| \left\| v_j \right\|_{L^2(\mathbb{P}_{N_0})} - \left\| v_j \right\|_{L^2(P_0)} \right|$$

Now the first term converges to zero by Hölder's inequality and the local uniform convergence of the optimal transport maps from above. The second term satisfies

$$\limsup_{N_0 \to \infty} \left| \left\| v_j \right\|_{L^2(\mathbb{P}_{N_0})} - \left\| v_j \right\|_{L^2(P_0)} \right|$$

$$= \limsup_{N_0 \to \infty} \left| \left( \int_{\mathbb{R}^d} \left| v_j(x_0) \right|^2 \mathrm{d}\mathbb{P}_{N_0} \right)^{1/2} - \left( \int_{\mathbb{R}^d} \left| v_j(x_0) \right|^2 \mathrm{d}P_0 \right)^{1/2} \right|$$

$$\leqslant \limsup_{N_0 \to \infty} \left| \int_{\mathbb{R}^d} \left| v_j(x_0) \right|^2 \mathrm{d}\mathbb{P}_{N_0} - \int_{\mathbb{R}^d} \left| v_j(x_0) \right|^2 \mathrm{d}P_0 \right|^{1/2}.$$

But since $P_0$ has finite second moments, it holds that this term also converges to zero.

Based on this we can show that $\widehat{\gamma}_{0j} \equiv \left( \mathrm{Id}, \widehat{b}_{0j} \right)$ converge weakly to $\gamma_{0j} \equiv (\mathrm{Id}, v_j)$. Indeed, if $\gamma_{0j}$ is a limit point of the sequence $\widehat{\gamma}_{0j}$, it holds that

$$\int_{(\mathbb{R}^d)^2} \left| x_j \right|^2 \mathrm{d}\gamma_{0j}(x_0, x_j) \leqslant \liminf_{N_0 \wedge N_j \to \infty} \int_{(\mathbb{R}^d)^2} \left| x_j \right|^2 \mathrm{d}\widehat{\gamma}_{0j}(x_0, x_j)$$

$$\leqslant \limsup_{N_0 \wedge N_j \to \infty} \int_{(\mathbb{R}^d)^2} \left| x_j \right|^2 \mathrm{d}\widehat{\gamma}_{0j}(x_0, x_j)$$

$$= \limsup_{N_0 \wedge N_j \to \infty} \int_{\mathbb{R}^d} \left| \widehat{b}_{0j}(x_0) \right|^2 \mathrm{d}\mathbb{P}_{N_0}(x_0)$$

$$= \int_{\mathbb{R}^d} \left| v_j(x_0) \right|^2 \mathrm{d}P_0(x_0).$$

Disintegrating the left-hand side with respect to $P_0$, and applying Jensen's inequality, gives

$$\int_{(\mathbb{R}^d)^2} \left| x_j \right|^2 \mathrm{d}\gamma_{0j}(x_0, x_j) = \int_{\mathbb{R}^d} \int_{\mathbb{R}^d} \left| x_j \right|^2 \mathrm{d}\gamma_{j|x_0}(x_j) \, \mathrm{d}P_0(x_0)$$

$$\geqslant \int_{\mathbb{R}^d} \left| \int_{\mathbb{R}^d} x_j \, \mathrm{d}\gamma_{j|x_0}(x_j) \right|^2 \mathrm{d}P_0(x_0)$$

$$= \int_{\mathbb{R}^d} |b_{0j}(x_0)|^2 \, \mathrm{d}P_0(x_0),$$

that is,

$$\int_{\mathbb{R}^d} |b_{0j}(x_0)|^2 \, \mathrm{d}P_0(x_0) \leqslant \int_{\mathbb{R}^d} |v_j(x_0)|^2 \, \mathrm{d}P_0(x_0).$$

But since $v_j$ is an optimal transport map between $P_0$ and $\widetilde{P}_j$ by definition, it holds that

$$\int_{\mathbb{R}^d} |b_{0j}(x_0)|^2 \, \mathrm{d}P_0(x_0) \geqslant \int_{\mathbb{R}^d} |v_j(x_0)|^2 \, \mathrm{d}P_0(x_0),$$

which implies that equality holds and we have that

$$\int_{\mathbb{R}^d} \left[ |b_{0j}(x_0)|^2 - |v_j(x_0)|^2 \right] \mathrm{d}P_0(x_0) = 0,$$

which implies that $b_{0j} = v_j$ $P_0$-almost everywhere. We have hence shown that $\left( \mathrm{Id}, \widehat{b}_{0j} \right)_{\#} \mathbb{P}_{N_0}$ converges weakly to $(\mathrm{Id}, b_{0j})_{\#} P_0$ for all $j$, where the barycentric projection $b_{0j}$ is the optimal transport map between $P_0$ and $\widetilde{P}_j$ (e.g. Villani, 2003, Theorem 2.12.(iii)).

Moreover, we have shown "strong $L^2$-convergence" of the barycentric projections in terms of Definition 5.4.3 in Ambrosio et al. (2008). Since this holds for all $j$, it also holds for their convex combination for fixed weights $\lambda \in \Delta^J$. Putting everything together, we then have that

$$\lim_{\bigwedge_j N_j \to \infty} \left\| \sum_{j=1}^{J} \lambda_j \widehat{b}_{0j} - \mathrm{Id} \right\|_{L^2(\mathbb{P}_{N_0})}^2 = \left\| \sum_{j=1}^{J} \lambda_j b_{0j} - \mathrm{Id} \right\|_{L^2(P_0)}^2 .$$

Since all observable measures $\mathbb{P}_j$ are empirical measures, they converge weakly in probability (Varadarajan, 1958), which implies that

$$\lim_{\bigwedge_j N_j \to \infty} P \left( \left| \left\| \sum_{j=1}^{J} \lambda_j \widehat{b}_{0j} - \mathrm{Id} \right\|_{L^2(\mathbb{P}_{N_0})}^2 - \left\| \sum_{j=1}^{J} \lambda_j b_{0j} - \mathrm{Id} \right\|_{L^2(P_0)}^2 \right| > \varepsilon \right) = 0 \qquad \text{for all } \varepsilon > 0.$$

This shows convergence in probability of the objective function for fixed $\lambda$.

***Step 2: Convergence of the optimal weights $\widehat{\lambda}_N^*$*** The convergence of the optimal weights now follows from standard consistency results in semiparametric estimation. In particular, the objective functions are all convex for any $\lambda \in \mathbb{R}^J$, which implies that they converge uniformly on any compact set (Rockafellar, 1970, Theorem 10.8), so the objective function converges uniformly on $\Delta^J$. Now a standard consistency result like Theorem 2.1 in Newey & McFadden (1994) then implies that

$$\lim_{\bigwedge_j N_j \to \infty} P \left( \left| \widehat{\lambda}_N^* - \lambda^* \right| > \varepsilon \right) = 0 \qquad \text{for all } \varepsilon > 0,$$

which is what we wanted to show. Note that the result can also be shown if we allow the weights $\lambda$ to be negative, i.e., if we only require that $\sum_{j=1}^J \lambda_j = 1$. In this case, the fact that the objective functions are convex and coercive implies that an optimal $\lambda^*$ will be achieved at the interior of the extended Euclidean space, from which consistency follows by Theorem 2.7 in Newey & McFadden (1994). ∎

*Proof of Corollary 3.1.* We want to show that $\left( \sum_{j=1}^J \widehat{\lambda}_{N_j}^* \widehat{b}_{0j} \right)_{\#} \mathbb{P}_{N_0}$ converges weakly in probability to $\left( \sum_{j=1}^J \lambda_j^* b_{0j} \right)_{\#} P_0$, where $\widehat{\lambda}_N^* \triangleq \left( \widehat{\lambda}_{N_1}^*, \ldots, \widehat{\lambda}_{N_J}^* \right)$ are the optimal weights obtained in

equation 3.1 and equation 2.6, respectively. The result follows by applying the extended continuous mapping theorem (van der Vaart & Wellner, 2013, Theorem 1.11.1) as follows.

As shown in the proof of Proposition 3.1 we have "strong $L^2$-convergence" of the maps $\sum_{j=1}^{J} \hat{\lambda}_{N_j}^* \hat{b}_{0j} - \text{Id}$ to $\sum_{j=1}^{J} \lambda_j^* b_{0j} - \text{Id}$. Therefore, by Theorem 5.4.4 (iii) in Ambrosio et al. (2008), it holds that

$$\lim_{\wedge_j N_j \to \infty} \int_{\mathbb{R}^d} f\left(x_0, \sum_{j=1}^{J} \hat{\lambda}_{N_j}^* \hat{b}_{0j}(x_0) - x_0\right) d\mathbb{P}_{N_0}(x_0) =$$

$$\int_{\mathbb{R}^d} f\left(x_0, \sum_{j=1}^{J} \lambda_j^* b_{0j}(x_0) - x_0\right) dP_0(x_0)$$

for any continuous function such that $|f(x_0)| \leqslant C_1 + C_2 |\overline{x}_0 - x_0|^2$ for all $x_0$ in the support of $P_0$, where $C_1, C_2 < +\infty$ are some constants and $\overline{x}_0$ in some element in the support of $P_0$ (Ambrosio et al., 2008, equation (5.1.21)). In particular, this holds for any bounded and continuous function $f$, which implies that

$$\lim_{\wedge_j N_j \to \infty} \int_{\mathbb{R}^d} f\left(\sum_{j=1}^{J} \hat{\lambda}_{N_j}^* \hat{b}_{0j}(x_0)\right) d\mathbb{P}_{N_0}(x_0) = \int_{\mathbb{R}^d} f\left(\sum_{j=1}^{J} \lambda_j^* b_{0j}(x_0)\right) dP_0(x_0)$$

for any bounded and continuous function, which implies that $\left(\sum_{j=1}^{J} \hat{\lambda}_{N_j}^* \hat{b}_{0j}\right)_{\#} \mathbb{P}_{N_0}$ converges weakly to $\left(\sum_{j=1}^{J} \lambda_j^* b_{0j}\right)_{\#} P_0$ if $\mathbb{P}_{N_j}$ converge weakly to $P_j$, $j \in \llbracket J \rrbracket$.

Now we apply the extended continuous mapping theorem (van der Vaart & Wellner, 2013, Theorem 1.11.1). Equip $\mathscr{P}_2(\mathbb{R}^d)$ with any metric $\widetilde{d}(\cdot, \cdot)$ that metrizes weak convergence. We define the maps $g : \bigtimes_{j=0}^{J} \left(\mathscr{P}_2(\mathbb{R}^d), \widetilde{d}\right)_j \to \left(\mathscr{P}_2(\mathbb{R}^d), \widetilde{d}\right)$ by

$$g(P_0, \ldots, P_J) = \left(\sum_{j=1}^{J} \lambda_j^* b_{0j}\right)_{\#} P_0 \,,$$

and analogously for their empirical counterparts $g_N$. Note that $g$ and $g_N$ are non-random functions if the measures $P_j$ and $\mathbb{P}_{N_j}$ are non-random themselves for all $j \in \llbracket J \rrbracket$. Moreover, by definition, $g$ and $g_N$ are continuous maps because $\sum_{j=1}^{J} \lambda_j^* b_{0j}$ are gradients of convex functions, which are continuous $P_0$-almost everywhere; the same thing holds for their empirical counterparts. Now from what we have shown above and in Proposition 3.1, it holds that

$$g_N\left(\mathbb{P}_{N_0}, \ldots, \mathbb{P}_{N_J}\right) \to g(P_0, \ldots, P_J)$$

as $\mathbb{P}_{N_j}$ converge weakly to $P_j$. Since $\{\mathbb{P}_{N_j}\}_{j=1}^{J}$ here instead are the only random elements in $\bigtimes_{j=0}^{J} \left(\mathscr{P}_2(\mathbb{R}^d), \widetilde{d}\right)_j$, the extended continuous mapping theorem implies that

$$\lim_{\wedge_j N_j \to \infty} P\left(\widetilde{d}\left(g_N\left(\mathbb{P}_{N_0}, \ldots, \mathbb{P}_{N_J}\right), g(P_0, \ldots, P_J)\right) > \varepsilon\right) = 0 \qquad \text{for all } \varepsilon > 0 \,,$$

which is what we wanted to show. ∎

## C   DETAILS OF MEDICAID EXPANSION APPLICATION

We use the ACS data with harmonized variables made available by IPUMS, a unified source of Census and survey data collected around the world. The data is at the household-person-year level. For our application, we select the household head and the spouse as our unit of analysis. The continuous outcomes are adjusted using the person-level sample weights available in the data.

We adopt the following sample restriction criteria: we included individuals

- of working age, i.e. between ages 18 and 65
- who has no missing outcomes (for those listed in the main text)
- who has no top-coded responses
- who are either household heads or their spouses

The sample size breakdown by states are follows:

| State | Observations |
|---|---|
| **Target** | |
| MT | 25,173 |
| **Control** | |
| AL | 106,464 |
| FL | 427,397 |
| GA | 227,659 |
| KS | 74,812 |
| MS | 61,505 |
| NC | 233,804 |
| SC | 107,905 |
| SD | 22,563 |
| TN | 152,470 |
| TX | 598,222 |
| WI | 157,410 |
| WY | 15,666 |

Table 1: Summary of the full data sample used to obtain $\lambda^*$.

We randomly select $N = 1500$ observations from each unit for estimating $\lambda^*$. In the Python implementation, we face a challenge where if the entries of the target and control data are large enough, equation A.3 becomes too large for `CVXPY` to compute an optimal solution. Therefore, we introduce a stabilizing constant to prevent this. This stabilizing constant is determined by the mean value and dimensions of the target distribution, and the number of controls. The weights we obtained are sparse and are displayed in Table 2.

| State | AL | FL | GA | KS | MS | NC | SC | SD | TN | TX | WI | WY |
|---|---|---|---|---|---|---|---|---|---|---|---|---|
| Weight | 0.184 | 0 | 0 | 0 | 0.174 | 0 | 0.010 | 0.513 | 0 | 0 | 0.119 | 0 |

Table 2: Estimated Weights for Control States.

We check whether the obtained weights are fit for creating synthetic Montana by examining if they well-approximate actual Montana in the pre-treatment period. As seen in Figures 6 and 7, our projection is similar to the actual data.

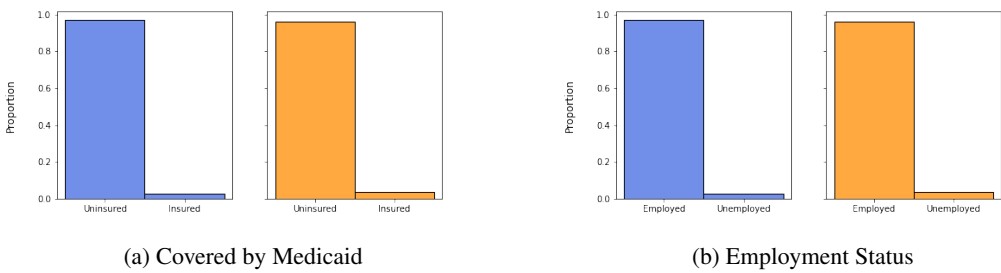

(a) Covered by Medicaid                    (b) Employment Status

Figure 6: Replicated (blue) vs actual (orange) Montana from 2010 to 2016.

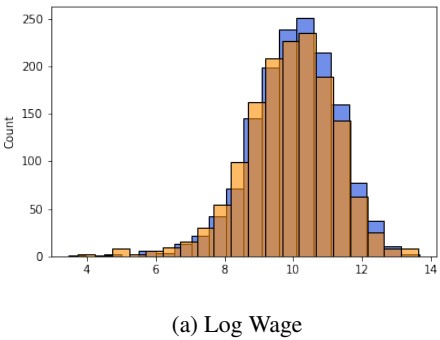 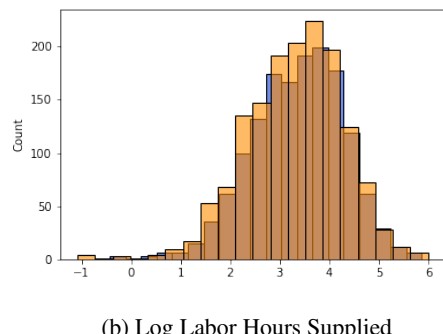

(a) Log Wage (b) Log Labor Hours Supplied

Figure 7: Replicated (blue) vs actual (orange) Montana from 2010 to 2016. In each panel, histograms of data distributions are shown on the left, and cumulative distribution functions are shown on the right.

Once we obtain the optimal weights $\lambda^*$, we estimate the counterfactual outcomes of interest for the four years after Medicaid expansion in Montana (namely, between 2017 and 2020). This involves solving equation A.1 with $\lambda^*$ obtained from the pre-intervention period. Implementation-wise, we computed the free-support barycenter, using the `POT` package; this does not fix the support of the barycenter *a priori*, and allows it to be different from those of the control distributions. We plot the densities and distributions of the counterfactual outcomes in Figure 5 of the main text.

To perform inference on the estimated causal effect, we use a placebo permutation test in analogy to Abadie et al. (2010), Gunsilius (2022). The idea is to repeatedly apply the procedure described above to each control unit, pretending in turn each control unit is the treated unit. Post-intervention, if an actual treatment effect only appears in the treatment unit (Montana, in this application), then the estimated effect for the actual treatment unit should be among the largest.

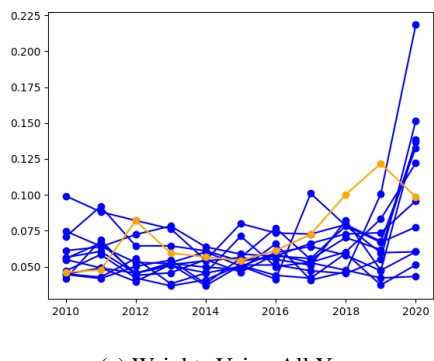 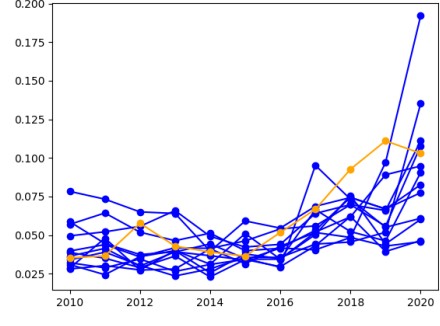

(a) Weights Using All Years (b) Weights Using Averaged Weights Over All Years

Figure 8: In orange: Montana. In blue: pretending each control state listed in Table 1 is a treated state.

We plot the 2-Wasserstein distance between the treated, joint distribution of *all outcomes* and the pre-/post-intervention optimal projection (i.e. equation A.1 with $\lambda^*$). We present two sets of results in Figure 8: in panel (A), the optimal projection is computed using $\lambda^*$ estimated using all years in the pre-intervention period; in panel (B), the $\lambda^*$ used is constructed from taking a simple average of weights estimated in each year of the pre-intervention period. Our results suggest that the estimated causal effect is valid in the post-intervention period, as we consistently observe the largest difference coming from Montana, especially from 2017-2019. The effect is less pronounced in 2020, however.

To accompany Figure 8, we also compute $p$-values, which we denote by and define as $p_t \triangleq \frac{r(d_{1t})}{J+1}$, where $d_{1t}$ is the 2-Wasserstein distance from the optimal projection to actual distribution when the target unit is Montana, $r(d_{1t})$ is the rank of $d_{1t}$ amongst $d_{jt}$s at given time $t$, and $J$ is the number of control units. Results are described in Table 3. A smaller $p_t$ value indicates larger treatment effect. We observed that $r(d_{1t}) = 1$ for 2018 and 2019, implying a nontrivial effect of the Medicaid expansion in Montana during these years. The values are $p_t$ are comparably higher in 2017 and 2020, which we attribute to the fact that it was the first year of the policy implementation, and the COVID-19 pandemic, respectively.

| Year ($t$) | $p_t$ (Weights Using All Years) | $p_t$ (Averaged Weights Over All Years) |
|---|---|---|
| 2017 | 0.231 | 0.308 |
| 2018 | 0.077 | 0.077 |
| 2019 | 0.077 | 0.077 |
| 2020 | 0.535 | 0.385 |

Table 3: Estimated $p_t \triangleq \frac{r(d_{1t})}{J+1}$ in the post-intervention period.

