# OpenReview forum: "Tangential Wasserstein Projections"
_ICLR.cc/2023/Conference — Submitted to ICLR 2023_

### Official Review · Reviewer_PFzW · 2022-10-15

**Confidence:** 4
**Correctness:** 3
**Technical Novelty And Significance:** 2
**Empirical Novelty And Significance:** 2
**Recommendation:** 3

**Clarity, Quality, Novelty And Reproducibility:**

The proposed idea is novel but I would argue that it is obvious and lies on the surface. Indeed, the theory and the entire idea presented here might seem non trivial, but actually it is just the
* linearization of the Wasserstein-2 space based on the Monge embedding w.r.t. P0 (which maps measures P to transport maps P0->P which are elements of the functional L2(P) space), see definition 1.1 in [1]
* Projection of the identity function “x” in L2(P) to the convex Hull of Monge embeddings of P1, P2,...,Pn which can be reduced to n-dimensional convex problem.

I do no understand why do the authors overcomplicate the entire exposition so much and why the theory of the tangential spaces is needed in this paper. It does not provide any useful information. In my view, everything what is described in pages 4-6 could and should be compressed to at most 1 page.

The experimental part is not sufficient to demonstrate that the proposed methodology is practically useful. The main issue here that it is not possible to understand the experimental setup of section 4. What are the distributions P_0, P_1,.., P_n here, how exactly are they obtained? It is not even written what is the dimension of the considered experiment. I bet some details are in Appendix, but this is not satisfactory as all the major details should be explicitly clarified in the main text. I think that it would be several times better if the authors include more experiments and experimental details to the main text instead of too long exposition of the tangential projection (see my comment above).

Finally, as far as I understand, no baselines are considered in the paper. I wonder why the authors do not test their method at least in some setups of [2] and do not compare with [2]. Specifically, it would be interesting to see how the proposed method works in the case of Figure 10 of [2].

**Replicability:** moderate. On the one hand, there are not a lot of clear technical details of the implementation in the paper. On the other hand, the code for all the experiments seems to be present in the provided supplementary.

**References**

[1] Mérigot, Q., Delalande, A., & Chazal, F. (2020, June). Quantitative stability of optimal transport maps and linearization of the 2-Wasserstein space. In International Conference on Artificial Intelligence and Statistics (pp. 3186-3196). PMLR.

[2] Bonneel, N., Peyré, G., & Cuturi, M. (2016). Wasserstein barycentric coordinates: histogram regression using optimal transport. ACM Trans. Graph., 35(4), 71-1.


**Details Of Ethics Concerns:**

-

**Strength And Weaknesses:**

**Strength**

* The proposed projection methodology is straightforward and does not require solving a bi-level optimization problem as, for example, in the barycentric coordinates approach [2].
* Theoretical consistency results for the weights recovered from empirical samples;

**Weaknesses**

* The proposed approach to projection method is rather obvious;
* The writing of the paper is not very good in many aspects and the paper is hard to parse:
* * **(a)** the current text overcomplicates the proposed idea too much;
* * **(b)** the actual contributions are not clearly stated;
* * **(c)** the experimental part is not transparent;

* Experimental support of the proposed ideas is not sufficient;


**Summary Of The Paper:**

The paper proposes a way to project a probability measure P0 onto the finite population P1, P2,..,Pn of the other measures. The projection idea is based on the weighted averaging of the optimal transport maps P0->Pn for the quadratic transport cost (the projection implies computing these weights for averaging). The authors prove statistical consistence results for the measures available by empirical samples and show the application to estimate causal effects via synthetic controls.


**Summary Of The Review:**

I do not think that the contribution of this paper is significant. In my opinion, the authors explain rather straightforward things by employing non-trivial facts such as tangential spaces, etc. At the same time, this seems to be absolutely unnecessary as the space they work in is just linear. Thus, I think the paper does not present any non-trivial results or ideas. Moreover, the writing should be improved/simplified. The only potentially positive and interesting result is the consistency theorem, but it is outweighed by the above-mentioned drawbacks of this paper. Thus, I think this paper is currently far below the ICLR paper standard and I vote for rejection.

---

> ### Author Response · Authors · 2022-11-18
> **Response to Reviewer PFzW**
>
> We thank you for your helpful review. Similar to the feedback from other reviewers, your review points to the fact that we used too much exposition in describing standard and well-known concepts and did not highlight our main contribution enough.
>
> We emphasize that our main contribution is a projection method that can be applied between general (i.e. not necessarily absolutely continuous) probability measures, while providing a global solution that can be obtained by a linear regression. In particular, it is based on a tangent cone structure, *not* the standard tangent space structure (i.e. Monge embedding) for absolutely continuous measures exploited in Mérigot et al. (2020); Werenski et al. (2022). This makes the exposition more complicated because we work with optimal transport *plans*, not maps. In the standard setting where the target measure is absolutely continuous with respect to Lebesgue measure, our approach reduces to the Monge embedding setting.
>
> We understand that the main contribution was not highlighted enough. The main changes we made to address this are as follows: we
>
> 1. condensed the exposition of theoretical background, related approaches, and special cases of our method from 4 to 2 pages. In so doing, we first relegated descriptions of the Wasserstein barycenter—which we used to place our contributions in the existing literature— to the appendix. We also relegated the special case of an absolutely continuous target measure to the appendix.
>
> 2. reorganized the application section of our approach to the synthetic controls method. In particular, we clarified the connection between constructing the synthetic controls estimator, and finding projections between sets of probability measures. We further explain the need for an extension to sets of general probability measures and a global and unique solution to the projection problem.
>
> 3. demonstrated another application to show how our approach compares to the standard Euclidean projection. We note that we were unable to locate the code used to create Figure 10 of Bonneel et al. (2016), especially since the authors changed the original 3D shape of the controls. We hence cannot perform one-to-one comparison between our approach and theirs on the same data. We have therefore included the results of an application of our method to the baseline benchmarks conducted in Werenski et al. (2022) and show that our method captures the nonlinearities of the problem better than the method in Werenski et al. (2022).
>
>
> **References**
>
> Bonneel, N., Peyré, G., and Cuturi, M. (2016). Wasserstein barycentric coordinates: histogram regression using optimal transport. ACM Transactions on Graphics, 35(4):1–10.
>
> Mérigot, Q., Delalande, A., and Chazal, F. (2020). Quantitative stability of optimal transport maps and linearization of the 2-wasserstein space. In International Conference on Artificial Intelligence and Statistics, pages 3186–3196. PMLR.
>
> Werenski, M. E., Jiang, R., Tasissa, A., Aeron, S., and Murphy, J. M. (2022). Measure estimation in the barycentric coding model. In International Conference on Machine Learning, pages 23781–23803. PMLR.

---

> > ### Comment · Reviewer_PFzW · 2022-11-23
> > **Response to the authors**
> >
> > Dear authors, I have read the rebuttal and looked through the updated paper. There are indeed some improvements in clarity and the overall exposition. However, I keep my score as is, see below.
> >
> > The current paper proposes a new **concept** for averaging probability distributions. As the current conference is about machine learning (rather than pure mathematics), as a reader and reviewer I expect to see some convincing application of the proposed concept. Thus, for this paper, I think it is mandatory to have a strong experimental part demonstrating that such *concept is indeed useful (?)*.
> >
> > Unfortunately, in my opinion, the experimental part in the current form (even after the revision) is not satisfactory. In the revised main text, there are two toy experiments and both are with only a single example per task (Figure 3/Figure 4). The Causal inference is (potentially) a promising one, but is poorly explained and it is hard to understand what has been done there. A lot of important details are moved to appendix (there are 3 references to appendix C in sec. 5!!!) which is not good. At the same time, the paper has a lot of empty space in the main text to wisely use it to clarify the necessary details more carefully. Beside, I think testing the proposed approach on a single dataset (sect. 5) is not a sufficient demonstration of its prospects.
> >
> > I do not understand how useful is the proposed tangential projections concept in practice. Due to this, the significance of theoretical results is questionable. I think this paper still requires a major revision with more thorough experimental evaluation.

---

> > > ### Author Response · Authors · 2022-11-28
> > > **Response to Reviewer PFzW (2nd Response)**
> > >
> > > Thank you for your response. We note that your main critique is our contribution is too mathematical, and the experimental section is too short (only containing two toy examples) to demonstrate the usefulness of our proposed method. Furthermore, you have reservations about the application to synthetic controls, which you thought is potentially interesting but not easy to understand.
> > >
> > > **To your first point**, we note that we have included four experiments in the main text:
> > >
> > > 1. a mixture of Gaussian examples in 10 dimensions, where we do not obtain sparse weights, as requested by another reviewer;
> > > 2. an application to the MNIST experiment in Werenski et al. (2022), showing that our method manages to replicate the non-linearities in the handwritten images compared to that of Werenski et al. (2022);
> > > 3. an application to images of Lego Bricks, which shows that our method manages to replicate the target well, even with only a few control measures (images). This application is a stand-in for the requested comparison to Bonneel et al. (2016), which we cannot perform, because it is not clear how the authors deformed 3D models to obtain their results;
> > > 4. An application to synthetic control methods, where we not only show that our method works for 28-dimensional non-regular measures, but also obtain novel results concerning the causal impact of Medicaid expansion in an adopter state. Please note that both of these
> > > points are made in the main text—the former in the introduction (end of Section 1.2), the latter in Section 5.
> > >
> > > **To your second point**,
> > >
> > > 1. you mentioned that a lot of details are moved to the appendix, and that we referred to the appendix 3 times. Please note that all referrals to the appendix are for auxiliary results about the actual Medicaid expansion and about the data, not for details of our mathematical
> > > and statistical contributions.
> > > 2. in the main text, we have explained all necessary details about our application to the synthetic controls method. In particular, we explained how the synthetic controls method is itself a projection, and how our proposed method applies to generalize it.
> > >
> > > **To your overarching point** about the practical relevance of our method, we want to reiterate that the application to synthetic controls is not possible with any other method because
> > > 1. the measures that we considered are not absolutely continuous w.r.t. Lebesgue measure (which rules out the methods by Werenski et al. (2022) and Mérigot et al. (2020));
> > > 2. the measures are 28-dimensional (which is challenging for Bonneel et al. (2016));
> > > 3. we obtain a unique global solution for our method in this setting, which rules out the implementation of Bonneel et al. (2016).
> > >
> > > We hope we have addressed all your comments, and we thank you again for your time.
> > >
> > > **References**
> > >
> > > Bonneel, N., Peyré, G., and Cuturi, M. (2016). Wasserstein barycentric coordinates: histogram regression using optimal transport. ACM Transactions on Graphics, 35(4):1–10.
> > >
> > > Mérigot, Q., Delalande, A., and Chazal, F. (2020). Quantitative stability of optimal transport maps and linearization of the 2-wasserstein space. In International Conference on Artificial Intelligence and Statistics, pages 3186–3196. PMLR.
> > >
> > > Werenski, M. E., Jiang, R., Tasissa, A., Aeron, S., and Murphy, J. M. (2022). Measure estimation in the barycentric coding model. In International Conference on Machine Learning, pages 23781–23803. PMLR

---

### Official Review · Reviewer_mwRK · 2022-10-23

**Confidence:** 2
**Correctness:** 3
**Technical Novelty And Significance:** 3
**Empirical Novelty And Significance:** 2
**Recommendation:** 6

**Clarity, Quality, Novelty And Reproducibility:**


(1) I feel that background and contribution should be split so that the contribution of the authors is much more clear. For example, Section 2.1 and the first half of Section 2.2 and Section 2.3 are backgrounds. While Eq.(2.3) and Section 2.4 onwards appear to be the author's contribution which appears to be mixed into the background. (of course, this is just the preference of the writing style, the authors may say this reads a bit better. But I tend to disagree.)

(2) Are the control measures the same size as the target? If so, how scalable is this? It doesn't appear to be scalable to high dimensions at all. This should be at least discussed as from my understanding, the projection into lower dimensional space is to make this much more computationally efficient. But the complexity of the model was not been discussed at all.

(3) Section 4 doesn't appear to be self-contained. After reading Section 4, I have no idea what you have done. I may need to read your references to understand your experiment setup.

**Strength And Weaknesses:**

Strengths
(1) Addresses an interesting research question by approximating a high-dimensional probability measure with low-dimensional settings using a geometric average called the barycentric projections.

Weaknesses
(2) See clarity


**Summary Of The Paper:**

The authors propose to use the 2-Wasserstein space over some Euclidean space to approximate high-dimensional probability measures in a low-dimensional setting. The authors proposed to do this in three steps, (i) the first is to obtain the general tangent cone structure at the target measure, (ii) the second is to construct a regular tangent space if it doesn't exist, and (iii) the third is to perform a linear regression to carry out the projection in the tangent space.

The challenge is to compute the corresponding optimal transport plan between the target and each measure used in the projection. The authors propose to do this using the classical synthetic control estimator. The introduction of the control estimate creates a hierarchical optimization, where first the distance is minimized between the parameters of the model and the set of control measures. The second layer of optimization then finds the minimum between the minimum of the set of control measures and the target (Eq.(2.3).

The authors show that the hierarchical structure is well defined between the regular target measure and the second of control measures and the target measure and the set of control measures. The authors also derive the consistency of the optimal weights/



**Summary Of The Review:**

Overall, the paper is well-written and easy to follow. I feel that the paper has some reasonable novelty to it. However, the main drawbacks are that there is no discussion of the computational complexity and a weak experiment section without any clarity of the experiment set up, and no benchmark to compare against to properly evaluate the performance of their proposed approach over other state-of-the-art approaches. Though, I do understand that this is a theoretical paper, but some sort of benchmark with comparably approaches should be made to show where it sits in the literature.

---

> ### Author Response · Authors · 2022-11-18
> **Response to Reviewer mwRK**
>
> We thank you for your helpful review. Here are our responses:
>
> 1. Regarding the point about clearly separating background from contributions, we agree that we had introduced too many concepts not directly related to our proposed approach. We did this to situate our method in the existing literature, but realized that this obfuscated our main contribution. To address this, we have
>  **i**. revised the introduction to distinguish between existing approaches in the literature and our contributions.
>  **ii**. focused on our main contribution, which is the projection method for general target measures $P_0$, and have relegated details that locate our approach in the existing literature— such as those of Wasserstein barycenters and the special case of an absolutely continuous target measure—to the appendix.
>
> 2. Indeed, the control measures are defined on the same ambient space, $\mathbb R^d$, as the target measure. The main source of computational and statistical complexity of our approach stems entirely from estimating the optimal transport plans since the subsequent projection in the tangent structure takes the form of a simple linear regression. We've now commented on this in our introduction: that the computational complexity of our approach scales with the complexity of estimating optimal transport plans. The complexity of computing optimal transport plans is well-known for many approaches; the nice part about our method is that it is compatible with essentially any approach to estimate the optimal transport plan, even regularized approaches, which is what we mention in the introduction.
>
> 3. We agree with the assessment about the point regarding Section 4 of original submission, on the application of our approach to the synthetic controls method. We've now reorganized the writing in the rebuttal submission. In particular, we
>     - described the connection between constructing the synthetic controls estimator, and finding projections between sets of probability measures;
>     - explained the need for an extension to sets of general probability measures, and a global (and unique) solution to the projection problem; and
>     - clarified the experiment setup, specifically the joint distribution of outcomes we considered, as well as the data sources used in the experiment.
>
> 4. We have included the results of an application of our method to the baseline benchmarks conducted in Werenski et al. (2022). We demonstrate how our approach compares to the standard Euclidean projection, in another illustration using Lego images.
>
>
> **Reference**
>
> Werenski, M. E., Jiang, R., Tasissa, A., Aeron, S., and Murphy, J. M. (2022). Measure estimation in the barycentric coding model. In International Conference on Machine Learning, pages 23781–23803. PMLR

---

### Official Review · Reviewer_LCUu · 2022-10-23

**Confidence:** 4
**Correctness:** 3
**Technical Novelty And Significance:** 3
**Empirical Novelty And Significance:** 2
**Recommendation:** 6

**Clarity, Quality, Novelty And Reproducibility:**

Clarity:
Overall the paper is weel written.

Quality:
The paper presents an interesting problem and develops interesting results related to it

Novelty:
The idea in (2.7) seems interesting but I am unclear about its relationship with (2.6)

Comments
1. In (2.6), the OT is between the barycenter and the control measures, whereas in (2.7) the OT is between the target P_0 and control measures. May be I am missing something, but is there a quantitative relation the provides the bounds the difference between these two solutions? Or is there any other justification that I am missing?
2. By the very nature, (2.7) promotes sparse optimal solutions. So will be performace be good enough when non-sparse combinations are actually involved? Even in the simulations, the application seems to have a sparse optimal solution. Can we have some simulation results on applications where non-sparse lambda is optimal?


**Strength And Weaknesses:**

Strength:
1. Paper is well written with enough references.
2. Prop 2.2 seems interesting,

Weakness:
1. The abstract reads that the proposed methodology has several applications. However, in simulations or otherwise only one application is shown. Also, this application does not seem to be one where the proposed method is compelling. In the sense that estimating causal effect has many solutions and the proposed seems only one of the ways. Are there any more compelling applications where the problem of finding the projection is more natural/compelling?

**Summary Of The Paper:**

The idea is to define an appropriate notion of W2 projection of P_0 onto given measures P_1,..,P_J . The proposed definition avoids the standard bi-level optimization based one and hence is efficient to compute. Simulations on a problem of estimating causal effect are presented.

**Summary Of The Review:**

Overall the paper is interesting, however, I currently see a couple of important gaps as summarized above. Hence I recommend accept with reservations.

---

> ### Author Response · Authors · 2022-11-18
> **Response to Reviewer LCUu**
>
> We thank you for your thoughtful review. Here are our responses to the issues you raised:
>
> 1. Regarding the point on our applications, we understand that the application to synthetic controls did not make it clear why our method is needed. We have addressed this in two ways:
>  **i**. We have now included the results of several illustrations of our method in section 4.
>    **ii**. We further clarified the connection between constructing the synthetic controls estimator and finding projections between sets of probability measures. We highlighted why it is important to allow for general measures, and to have global (and unique) solutions. This is why we believe our application to synthetic controls method is compelling. These points are now made in Section 5.
>
>
> 2. With respect to the first point about the clarity of the exposition and notation, we agree that we introduced too many concepts peripheral to our approach, specifically that of the Wasserstein barycenter and the special case of a regular target measure. We have relegated both to the appendix.
>  We also want to clarify that equation (2.7) relates to our projection approach, where the optimal transport maps are defined between the target $P_0$ and controls $P_j$. In contrast, equation (2.6) describes the Wasserstein barycenter in the regular case, where the optimal transport maps are defined between the barycenter and controls. Note that the barycenter approach is not fundamental to our method. We merely described equation (2.6) and the related concepts in our original submission to place our contributions in the existing literature. We realize that this obscured our main contribution. To highlight our approach, we now focus exclusively on the case of a general target measure.
>
> 3. The sparsity observed follows from the fact that the target measure lies outside the generalized geodesic convex hull of the control measures. By definition a projection onto this convex hull will be achieved on one of the faces of the convex set, which by definition will set weights of control measures not part of this face to zero. It seems like in most practical setting the target lies outside the generalized geodesic convex hull (something also observed but not explained in Bonneel et al. (2016)).
>  To make this clearer, we have mentioned this in the main text. Moreover, we have also included an artificial simulation where the target does not lie outside the geodesic convex hull (Gaussian mixtures in section 4.1). Note that our method does not provide sparse weights in this case.
>
> **References**
>
> Bonneel, N., Peyré, G., and Cuturi, M. (2016). Wasserstein barycentric coordinates: histogram regression using optimal transport. ACM Transactions on Graphics, 35(4):1–10.

---

### Official Review · Reviewer_pNhL · 2022-10-26

**Confidence:** 3
**Correctness:** 3
**Technical Novelty And Significance:** 3
**Empirical Novelty And Significance:** 3
**Recommendation:** 6

**Clarity, Quality, Novelty And Reproducibility:**

The proposed idea is interesting and may have promising potential for further applications. However, the authors may need to improve the notations and discuss relations between the proposed method with the existing ones in the literature.

Some concerns are as follows:
+ For Section 2.4, the notations are confusing, the authors use the same notation \Delta \phi_j for Equ. (2.6) and (2.7), but with different meanings. What is the relation between them? Could the authors elaboration motivations, relations for changing the optimal map between barycenter and P_j into the one between P_0 and P_j?
---> In this case, one may not need to compute barycenter anymore for Problem 2.3, but it may simply precompute the optimal map between P_0 and P_j for every j = 1..J. Therefore, this relaxation simplify the Problem 2.3. a lot. It is better in case the authors elaborate their relation.

+ It is hard to follow the idea in Section 2.3. In Equ. (4), to define a geometric tangent cone structure at each measure P, could the authors elaborate what is the "optimal" meaning for (\pi_1, \pi_1 + \epsilon \pi_2)_# \gamma is "optimal" for some \epsilon > 0)?
--- Especially, in Equ. (2.4) it seems that the authors use optimal map to represent for each distribution (e.g., P_2, P_3) to the anchor P_0. It is better to discuss such choices with existing results in the literature.

Ref:
[For optimal map representation] Ref: Mérigot, Q., Delalande, A. and Chazal, F., 2020, June. Quantitative stability of optimal transport maps and linearization of the 2-Wasserstein space. In International Conference on Artificial Intelligence and Statistics (pp. 3186-3196). PMLR.

[For cone structure of W_2] Takatsu, A. and Yokota, T., 2012. Cone structure of L2-Wasserstein spaces. Journal of Topology and Analysis, 4(02), pp.237-253.


**Strength And Weaknesses:**

Strength:
+ The authors give a new notions for projection of a distribution into a set of distributions by leveraging regular tangent cones of the Wasserstein space.

Weaknesses:
+ The authors need to give motivation and elaborate with more details relations between the proposed approach with the existing one (e.g., Problem 2.6 and Problem 2.7 where the authors use the same notations, \Delta \phi_j but with different meaning, where in (2.6), it is the optimal transport map from the barycenter to the respective measure P_j while in (2.7), it is the optimal map between the target P_0 and P_j)
+ The authors need to discuss the relation between the propose method and the approach which use the optimal map to represent distributions (w.r.t. an anchor distribution, e.g., P_0 as in the submission),

Ref: Mérigot, Q., Delalande, A. and Chazal, F., 2020, June. Quantitative stability of optimal transport maps and linearization of the 2-Wasserstein space. In International Conference on Artificial Intelligence and Statistics (pp. 3186-3196). PMLR.


**Summary Of The Paper:**

The authors propose a new notion of projections of a distribution into a set of distributions with Wasserstein metric by working on regular tangent cones of the Wasserstein space using generalized geodesics. The authors evaluate the proposed method to estimate causal effects via synthetic controls.

**Summary Of The Review:**

The proposed idea is interesting and has a promising potential for applications. However, the authors need to elaborate more details about the proposed approach and its motivation/relation with the ones in the literature.

---

> ### Author Response · Authors · 2022-11-18
> **Response to Reviewer pNhL**
>
> We thank you for your detailed and thoughtful responses.
>
> 1. With respect to the clarity of the exposition and notation, we agree that we introduced too many concepts peripheral to our approach, especially those related to the Wasserstein barycenter. We relegated these concepts, which we used to connect our contributions to the existing literature, to section A in the appendix.
>  To clarify, equation (2.6) indeed describes the Wasserstein barycenter in the regular case, where the optimal transport maps are defined between the barycenter and controls. In contrast, equation (2.7) relates to the projection approach, where the optimal transport maps are defined between the target $P_0$ and controls $P_j$. We indeed overloaded the notations in our original writing; we've addressed this in the revision.
>
> 2. Thank you for pointing out the relation of our approach to Mérigot et al. (2020). We note that the tangential structure described therein is anchored at an absolutely continuous measure. In comparison, we do not require absolute continuity on any of the measures we consider. We have made this point clear in the revision.
>
> 3. As for the first of your other concerns, it is correct that the Wasserstein barycenter is not directly related to our approach. We mentioned this to place our approach in the existing related literature. We have now placed all these peripheral exposition in the appendix, as noted above.
>
> 4. As for the definitions in the old Section 2.3: in words, $\mathcal G(P)$ is defined as the collection of all couplings between $P$ and any other measures, in such a way that they are optimal transport plans for at least a small "distance" away from $P$. In the special case where the measure $P$ is absolutely continuous, the tangent cone structure $\mathcal G(P)$ reduces to the regular tangent space, as  discussed in Ambrosio et al. (2008, chapter 12).
>
>
> **References**
>
> Ambrosio, L., Gigli, N., and Savaré, G. (2008). *Gradient flows in metric spaces and in the space of probability measures*. Lectures in mathematics ETH Zürich. Birkhäuser, Basel, 2. ed edition.
>
> Mérigot, Q., Delalande, A., and Chazal, F. (2020). Quantitative stability of optimal transport maps and linearization of the 2-wasserstein space. In International Conference on Artificial Intelligence and Statistics, pages 3186–3196. PMLR.

---

### Author Response · Authors · 2022-11-18
**General Response**

We thank the reviewers for their thoughtful and helpful comments, in particular about the clarity of the exposition, and our contributions to the existing related literature—both of which were common themes throughout the reviews. Our contributions are: 1) allowing for general target measures in the projection, 2) providing a simple implementation via regressions, which 3) admit global—and in many cases, unique—solutions.

To streamline our exposition and to better highlight our contributions, we

1. revised the introduction to clearly delineate the existing approaches in the literature and our contributions.

2. relegated the extraneous exposition on Wasserstein barycenter and the case of regular target measure in the projection method—related to the approaches in Mérigot et al. (2020); Werenski et al. (2022)—to the appendix.

3. made the exposition in Section 2 on our method more concise. We now exclusively focus on the general case.

4. included results from our method performed on the benchmark described in Werenski et al. (2022). We further added simulation results from using mixtures of Gaussians and Lego brick images.

5. made clear how the synthetic controls method relates to finding projections between probability measures, why it is important to allow for general measures, and why it is important to have global and unique solutions. This is now contained in Section 5.


**References**

Mérigot, Q., Delalande, A., and Chazal, F. (2020). Quantitative stability of optimal transport maps and linearization of the 2-wasserstein space. In International Conference on Artificial Intelligence and Statistics, pages 3186–3196. PMLR.

Werenski, M. E., Jiang, R., Tasissa, A., Aeron, S., and Murphy, J. M. (2022). Measure estimation in the barycentric coding model. In International Conference on Machine Learning, pages 23781–23803. PMLR.

---

### Decision · Program_Chairs · 2023-01-20

**Decision:**

Reject

**Justification For Why Not Higher Score:**

Novelty, experiments.

**Justification For Why Not Lower Score:**

na

**Metareview: Summary, Strengths And Weaknesses:**

The paper proposes a new approach to project a measures onto a "linear" combination of probability measure, in the W2 space, a task which can be related to a "regression" problem in the Wasserstein space. Two approaches have been proposed for that task, either a bilevel optimization problem (where the barycenter of a set of measures using a family of weights is the argmin of a given energy, and its distance to the query point is further differentiated) or using some form of Linearization on the tangent cone of a reference measure to fall back on a Euclidean parameterization in which the barycentric problem is simply a (Euclidean) regression problem. Both approaches have been proposed. The novelty here, as underlined by the authors, is that the method uses, instead of building on Monge maps to a reference measure (which are difficult to access), the barycentric projections from similar plans as an approximation.

Reviewers have had split opinions on this paper. Ultimately, and after careful reading of their opinions and the author rebuttals, I propose to reject the paper based on a few issues. The first is that the paper has undergone significant changes during the review process, correcting for some flaws, which, as even positive reviewers aknowledge, should warrant a new round of fresh reviews due to their breadth. The second is that the experimental section remains weak, and one reviewer in particular has raised numerous concerns. Finally, the authors mention that

"In contrast, approaches that depend on the classical Wasserstein barycenter consider optimal transport maps between the barycenter and the control measures (e.g. Bonneel et al. (2016); Werenski et al. (2022))."

but as presented in the paper that distinction is only "temporary", since the authors naturally end up using a Monge map approximator, the barycentric projection operator

"To your second point on the sizes of optimal transport plans, we note that we apply the barycentric projection to the optimal transport plans, i.e.  in our notation. In particular, this means we always work in the same dimension as our target probability measure ."

However, https://arxiv.org/pdf/2201.12195.pdf (Sec.3.1) follow exactly the same route (including entropic regularization) since they rely on an entropic map estimator (identity - 1/2 gradient of dual potential) which is conceptually equivalent to what is proposed here.

As a result, it feels that the gap between the method proposed here and the method by Werenski (2022) needs to be clarified. That paper was published on arxiv close a year ago (ICML submission) and so more care must be used to differentiate this submission from that paper which looks very similar.